# Antibiotic-induced accumulation of lipid II synergizes with antimicrobial fatty acids to eradicate bacterial populations

Ashelyn E Sidders[1], Katarzyna M Kedziora[2,3], Melina Arts[4], Jan-Martin Daniel[4], Stefania de Benedetti[4], Jenna E Beam[1], Duyen T Bui[1], Joshua B Parsons[1,5], Tanja Schneider[4], Sarah E Rowe[1], Brian P Conlon[1,6]*

[1]Department of Microbiology and Immunology, University of North Carolina at Chapel Hill, Chapel Hill, United States; [2]Department of Genetics, University of North Carolina at Chapel Hill, Chapel Hill, United States; [3]Bioinformatics and Analytics Research Collaborative, University of North Carolina at Chapel Hill, Chapel Hill, United States; [4]Institute for Pharmaceutical Microbiology, University of Bonn, Bonn, Germany; [5]Division of Infectious Diseases, Duke University, Durham, United States; [6]Marsico Lung Institute, University of North Carolina at Chapel Hill, Chapel Hill, United States

**Abstract** Antibiotic tolerance and antibiotic resistance are the two major obstacles to the efficient and reliable treatment of bacterial infections. Identifying antibiotic adjuvants that sensitize resistant and tolerant bacteria to antibiotic killing may lead to the development of superior treatments with improved outcomes. Vancomycin, a lipid II inhibitor, is a frontline antibiotic for treating methicillin-resistant *Staphylococcus aureus* and other Gram-positive bacterial infections. However, vancomycin use has led to the increasing prevalence of bacterial strains with reduced susceptibility to vancomycin. Here, we show that unsaturated fatty acids act as potent vancomycin adjuvants to rapidly kill a range of Gram-positive bacteria, including vancomycin-tolerant and resistant populations. The synergistic bactericidal activity relies on the accumulation of membrane-bound cell wall intermediates that generate large fluid patches in the membrane leading to protein delocalization, aberrant septal formation, and loss of membrane integrity. Our findings provide a natural therapeutic option that enhances vancomycin activity against difficult-to-treat pathogens, and the underlying mechanism may be further exploited to develop antimicrobials that target recalcitrant infection.

**\*For correspondence:**
bconlon@med.unc.edu

## Editor's evaluation

The authors present how unsaturated fatty acids modulate the bactericidal effect of the antibiotic vancomycin. The authors find that palmitoleic acid significantly increases the bactericidal activity of vancomycin and reveal the mechanism responsible. The key findings will be of interest to a broad audience of researchers focused on microbiology, host-pathogen interactions, and antimicrobial development, as well as to clinicians that treat antibiotic-recalcitrant infections.

## Introduction

The 1950s were the golden age of antibiotic discovery with nearly half of our current arsenal of therapeutics discovered in a single decade (*Aminov, 2010*). Vancomycin, derived from the word 'vanquish,' is one of the key discoveries of the golden age that has maintained clinical utility through the present

day (*Rubinstein and Keynan, 2014*). Vancomycin is a glycopeptide antibiotic that targets lipid II, an essential membrane-bound building block of the cell wall. Lipid II is synthesized in the cytoplasm and subsequently flipped to the outer leaflet of the cell membrane where it is incorporated into the mature peptidoglycan layer (*Rajagopal and Walker, 2017*). Vancomycin binds to the terminal D-Ala-D-Ala residues of lipid II, sterically hindering its incorporation into the growing peptidoglycan layer, resulting in cessation of cell wall biosynthesis and subsequent accumulation of lipid II on the cell surface (*Grein et al., 2019*; *Qiao et al., 2017*). Since lipid II is highly conserved across bacterial species, it is often referred to as the 'Achilles' heel' of cell wall biosynthesis and an ideal antibacterial target (*Müller et al., 2017*).

Resistance to vancomycin is a growing problem that limits its utility against some infections (*De Oliveira et al., 2020*). Resistance is mediated by the acquisition of an inducible *van* cassette encoding seven proteins that generate an alternative lipid II peptide terminus, D-Ala-D-Lac, that significantly reduces vancomycin binding (*De Oliveira et al., 2020*). About 1 in 3 hospital-associated *Enterococcus faecalis* isolates are vancomycin-resistant. Such high-level vancomycin resistance is also possible in *Staphylococcus aureus*, although it is rare with only 14 cases identified to date in the United States (*McGuinness et al., 2017*). However, vancomycin-intermediate *S. aureus* (VISA) strains, which exhibit reduced susceptibility to vancomycin through a variety of non-specific adaptations including a thickening of the cell wall, are increasingly common (*McGuinness et al., 2017*). It is also important to acknowledge that vancomycin treatment frequently fails in the absence of resistance, and this failure is often attributed to antibiotic tolerance (*Van den Bergh et al., 2017*). Antibiotic tolerance is the ability of a bacterial population, or a sub-population of tolerant cells called persisters, to survive high concentrations of bactericidal antibiotics and regrow when antibiotic pressure is removed, leading to re-establishment of the infection (*Hill et al., 2021*). Additionally, antibiotic tolerance can lead to the evolution of resistance (*Van den Bergh et al., 2017*; *Levin-Reisman et al., 2017*). Thus, to improve patient outcome, we need to develop strategies that can target both antibiotic resistant and antibiotic tolerant populations.

Despite renewed interest in funding antibiotic development, the pipeline is sparse, with few new drugs reaching the market (*Lewis, 2020*). Developing feasible strategies to revitalize our current arsenal of antibiotics, such as the use of antibiotic adjuvants, which enhance the efficacy of an already approved therapeutic, are highly desirable (*Gray and Wenzel, 2020a*). Cell membrane-acting agents (CMAAs) have frequently been overlooked in drug discovery, as they often lack selectivity for bacterial membranes and display promiscuous activity toward mammalian cells (*Hurdle et al., 2011*). However, the clinical success and specificity of approved membraneactive antibiotics such as daptomycin, polymyxin, and second-generation glycopeptides have highlighted their utility (*Lewis, 2020*). Additionally, there are several membrane-targeting antibiotic candidates in pre-clinical development (*Lewis, 2020*; *Martin et al., 2020*; *Kim et al., 2019*). Our group and others have also previously identified CMAAs that significantly improved the efficacy of aminoglycosides against antibiotic-tolerant *S. aureus* (*Radlinski et al., 2017*; *Radlinski et al., 2019*; *Kim et al., 2018*; *Song et al., 2020*; *Beavers et al., 2022*).

Second-generation glycopeptides, known as lipoglycopeptides, are more potent variants of vancomycin that can bypass vancomycin resistance. Their improved activity is attributed to the lipophilic side chain of lipoglycopeptides that anchor into and disturb the bacterial membrane (*Müller et al., 2017*). Thus, we were interested in examining the ability of various CMAAs to potentiate vancomycin activity. Here, we identify two antimicrobial fatty acids that are potent vancomycin adjuvants against tolerant and resistant Gram-positive bacteria. Furthermore, we describe the mechanism by which these fatty acids and vancomycin work together to compromise the bacterial cell membrane, leading to rapid cell death.

## Results
### Unsaturated fatty acids potentiate vancomycin killing of *S. aureus*

We have previously shown that rhamnolipids, a biosurfactant produced by *Pseudomonas aeruginosa*, synergizes with aminoglycosides, highlighting the potential of targeting the membrane to enhance antibiotic efficacy against *S. aureus* (*Radlinski et al., 2019*). We reasoned that CMAAs may also enhance the killing activity of glycopeptides since second-generation glycopeptides have a lipophilic

tail and display potent activity (*Blaskovich et al., 2018*). We investigated the capacity of seven CMAAs to enhance the bactericidal activity of vancomycin (*Figure 1A*, *Figure 1—figure supplement 1*). Sublethal concentrations of each CMAA were established experimentally or from our previously published work (*Figure 1—figure supplement 2*; *Radlinski et al., 2019*). Vancomycin and CMAAs both have potent bactericidal activity against low-density bacterial populations ($10^5$–$10^6$ colony forming unit [CFU]/ml), but bactericidal activity wanes as population density increases (*LaPlante and Rybak, 2004*; *Kollef, 2007*; *Parsons et al., 2012*; *Loffredo et al., 2021*). We examined the bactericidal activity of vancomycin and CMAAs alone or in combination against *S. aureus* population of $10^8$ CFU/ml, similar to the cell density of an abscess or infected wound (*Pletzer and Hancock, 2018*; *König et al., 1998*; *Udekwu et al., 2009*). At this density, vancomycin monotherapy resulted in an approximately 2-log reduction in bacterial abundance but did not eradicate the population (*Figure 1A*). Benzyl alcohol, a general membrane fluidizer known to lack antimicrobial activity alone, did not display synergy with vancomycin (*Figure 1A*; *Parsons et al., 2012*; *Müller et al., 2016*). Lauric acid, a saturated fatty acid, and its monoglyceride derivative, glycerol monolaurate, which synergizes with other antibiotics such as aminoglycosides, did not synergize with vancomycin (*Figure 1A*; *Radlinski et al., 2019*; *Churchward et al., 2018*; *Hess et al., 2014*; *Fischer, 2020*). Adarotene, a synthetic retinoid that also potentiates aminoglycosides, did not potentiate vancomycin killing (*Figure 1A*; *Kim et al., 2018*). In contrast, rhamnolipids improved vancomycin killing (*Figure 1A*), while palmitoleic acid and linoleic acid, two host-produced unsaturated fatty acids (UFAs), were the most potent adjuvants tested, resulting in a 6-log decrease in viable bacteria after 6 hr (*Figure 1A*). Palmitoleic and linoleic acid are both *cis* UFAs, a 16-carbon monounsaturated and 18-carbon poly-unsaturated fatty acid, respectively (*Parsons et al., 2012*). Both UFAs have well-established anti-microbial activity and contribute to innate immunity in human nasal secretions, sebaceous glands of the skin, and breast milk (*Wille and Kydonieus, 2003*; *Do et al., 2008*). At the bacterial density and concentration examined here, neither palmitoleic acid nor linoleic acid had any activity against *S. aureus* in the absence of vancomycin (*Figure 1A*). However, when these UFAs were combined with vancomycin, we observed over 99% killing of both methicillin-sensitive *S. aureus* (MSSA) and methicillin-resistant *S. aureus* (MRSA) populations after only 30 min (*Figure 1B–C*). Importantly, the dual treatment eradicated the antibiotic-tolerant persister populations as indicated by the absence of a characteristic persister plateau (*Figure 1B–C*; *Sidders et al., 2021*). Additionally, we evaluated if dual treatment could bypass chemically induced tolerance using carbonyl cyanide m-chlorophenyl hydrazone (CCCP) to disrupt proton motive force or arsenate (AsKO) to deplete ATP (*Conlon et al., 2016*). While reduced ATP levels typically protect from antibiotics, we found that palmitoleic acid still potentiated vancomycin killing of *S. aureus* in a low energy, tolerant state (*Figure 1—figure supplement 3*). These data indicate that palmitoleic acid is a potent adjuvant against tolerant and susceptible *S. aureus* populations.

## Antibiotic synergy is mediated by the accumulation of membrane-bound cell wall intermediates

Because palmitoleic acid exhibited the greatest synergy with vancomycin, we next evaluated whether palmitoleic acid synergizes with other cell wall inhibitors. Antibiotics that target cell wall biosynthesis can be divided into three categories based on their target: early cytoplasmic steps (fosfomycin), membrane-bound steps (vancomycin and bacitracin), and assembly/incorporation steps of peptidoglycan (β-lactams; *Rajagopal and Walker, 2017*). The latter two categories make up the majority of clinically relevant antibiotics that target the cell wall (*Rajagopal and Walker, 2017*). To gain insight into the mechanism of synergy within the context of the peptidoglycan biosynthesis pathway, we evaluated representative cell wall-acting antibiotics from each category in combination with palmitoleic acid (*Figure 2A*). Fosfomycin inhibits the cytosolic steps of peptidoglycan biosynthesis by targeting and irreversibly inhibiting enzymes MurA-B (*Falagas et al., 2016*). Fosfomycin did not exhibit synergy with palmitoleic acid (*Figure 2B*). β-lactam antibiotics target penicillin-binding proteins (PBPs) which are responsible for incorporating lipid II into the mature peptidoglycan and the cross-linking of peptidoglycan (*Schneider and Sahl, 2010*). Oxacillin inhibits PBP2 while nafcillin inhibits all four PBPs in *S. aureus*. Neither β-lactams exhibit synergy with palmitoleic acid (*Figure 2C–D*). These data indicate that UFAs do not synergize with antibiotics that target the early cytoplasmic steps or the assembly/incorporation steps of peptidoglycan synthesis.

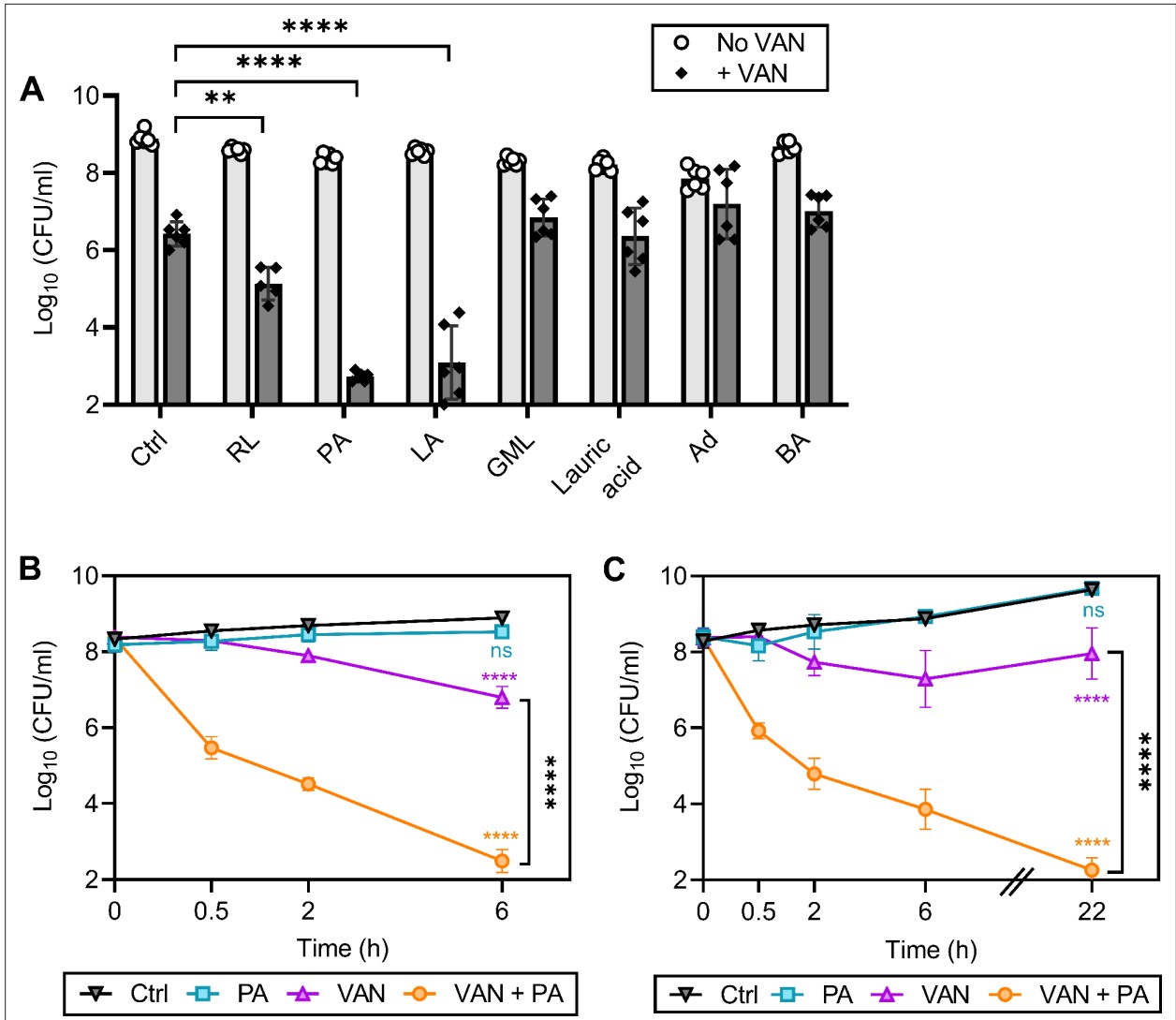

**Figure 1.** Palmitoleic acid rapidly potentiates vancomycin killing of *S. aureus*. (**A**) *S. aureus* HG003 cultures were grown to exponential phase and subsequently treated with CMAAs ± vancomycin (20 µg/ml, 20 X MIC of HG003). Colony forming units (CFUs) were enumerated after 6 hr. CMAAs tested include rhamnolipids (30 µg/ml), palmitoleic acid (11 µg/ml), linoleic acid (12 µg/ml), glycerol monolaurate (30 µg/ml), lauric acid (30 µg/ml), adarotene (3.2 µg/ml), and benzyl alcohol (40 mM). The statistical significance was determined using one-way ANOVA with Dunnett's multiple comparison test comparing dual treated conditions to vancomycin alone. Unless indicated, comparisons were not significant. Data represent n=6 biologically independent replicates. (**B**) Methicillin-sensitive *S. aureus* (MSSA) strain HG003 (**C**) or community-acquired methicillin-resistant *S. aureus* (MRSA) strain LAC was challenged with DMSO (Ctrl), palmitoleic acid (11 µg/ml), vancomycin (20 µg/ml), orcombination therapy. CFU was enumerated at indicated time points. Data represent the mean values from n=6 biologically independent replicates ± SD. Statistical significance was determined using a two-way ANOVA with Dunnett's multiple comparison test on the final time point evaluated (**B and C**). Comparisons made between the control and experimental conditions are indicated in the associated line color. ** and **** denote p<0.01 and p<0.0001, respectively. VAN, vancomycin; RL, rhamnolipids; PA, palmitoleic acid; LA, linoleic acid; GML, glycerol monolaurate; Ad, adarotene; BA, benzyl alcohol.

The online version of this article includes the following source data and figure supplement(s) for figure 1:

**Source data 1.** Related to *Figure 1A–C*.

**Figure supplement 1.** Structures of CMAAs evaluated for synergy with vancomycin.

**Figure supplement 2.** Identification of sublethal concentration of UFAs.

**Figure supplement 2—source data 1.** Related to *Figure 1—figure supplement 2A–D*.

**Figure supplement 3.** Palmitoleic acid potentiates vancomycin killing against chemically induced tolerance.

**Figure supplement 3—source data 1.** Related to *Figure 1—figure supplement 3A–B*.

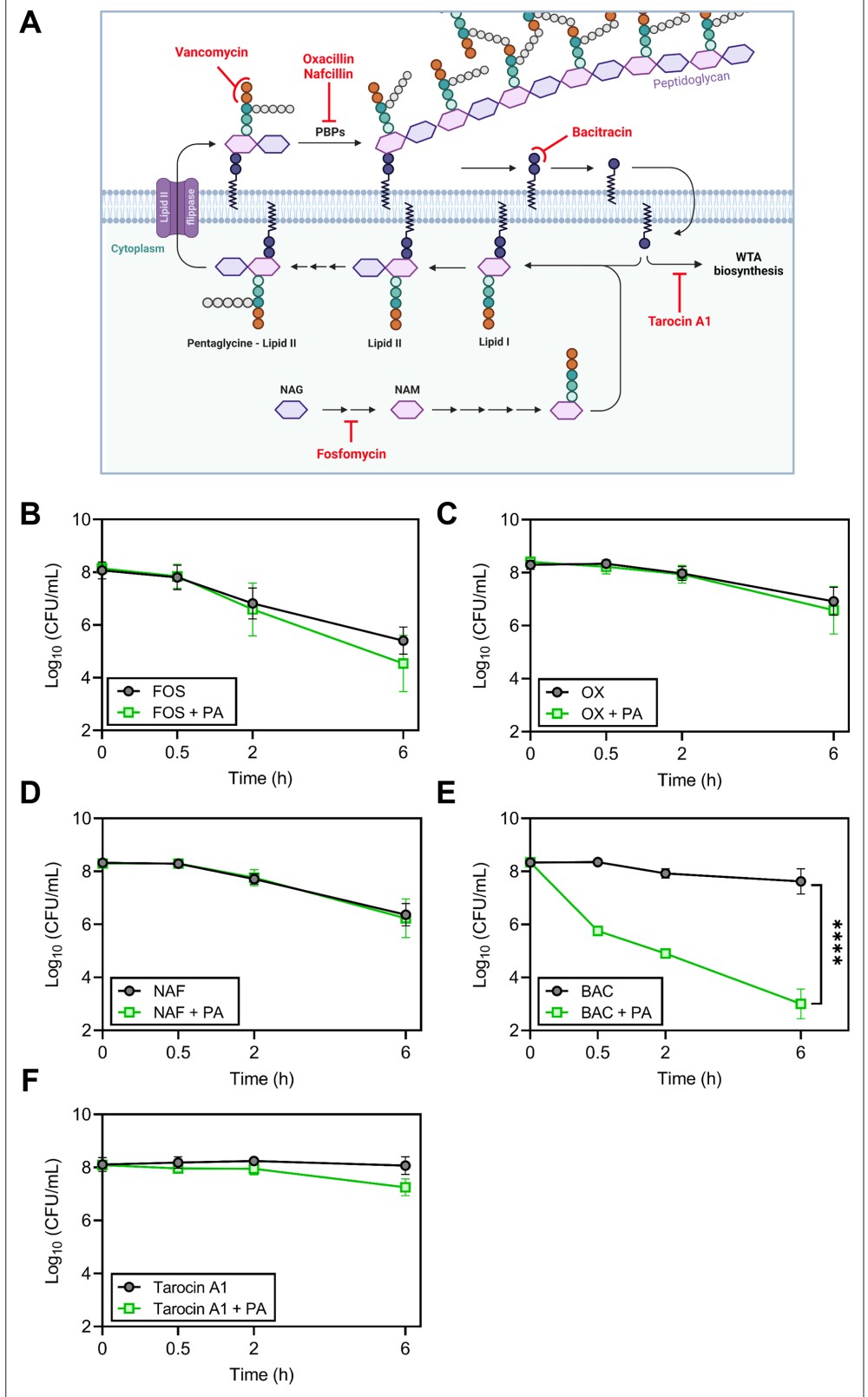

**Figure 2.** Accumulation of bactoprenol is necessary for palmitoleic acid potentiation of antibiotics. (**A**) Schematic depicting cell wall biosynthesis, with each arrow in between intermediates indicating an enzymatic step. The cell wall is composed mainly of peptidoglycan, a polymer consisting of N-acetyl-glucosamine (NAG) and N-acetylmuramic acid (NAM) residues, the latter affixed with a short pentapeptide. NAM is attached to a lipid

*Figure 2 continued on next page*

*Figure 2 continued*

anchor known as bactoprenol and further modified to generate lipid II. Lipid II is flipped to the outer leaflet of the membrane and incorporated into the growing peptidoglycan layer. Bactoprenol is recycled and reused in a cyclic process. The targets inhibited by the antibiotics tested in (**B–F**) are indicated in red. Schematic created with BioRender.com. (**B–F**) Survival of HG003 challenged with indicated antibiotic monotherapy (black circles) or the antibiotic combined with PA (11 µg/ml, green square). Colony forming units (CFUs) were enumerated at indicated time points. Antibiotics tested include (**B**): fosfomycin (250 µg/ml)±PA, (**C**): the PBP2 inhibitor, oxacillin (OX, 5 X MIC, 5 µg/ml)±PA, (**D**): the PBP1-4 inhibitor, nafcillin (NAF, 5 X MIC, 2.5 µg/ml)±PA, (**E**): bacitracin (BAC, 250 µg/ml)±PA, and (**F**): tarocin A1 (8 µg/ml)±PA. Data represent the mean values from n=6 biologically independent replicates ± SD. Statistical analysis was evaluated at the end point by a two-tailed unpaired Student's *t*-test with a 95% CI; conditions with significance are indicated on the graph, ****$p < 0.0001$, otherwise comparisons were not significant.

The online version of this article includes the following source data for figure 2:

**Source data 1.** Related to *Figure 2B–F*.

Vancomycin binds to the D-Ala-D-Ala moiety of lipid II and sterically hinders its incorporation into mature peptidoglycan; this indirectly causes an accumulation of lipid II on the cell membrane (*Qiao et al., 2017*). With this in mind, we wanted to determine if synergy with palmitoleic acid requires the accumulation of membrane-bound precursors. Bacitracin inhibits the recycling of the essential bactoprenol lipid carrier (*Figure 2A*; *Stone and Strominger, 1971*). Interestingly, while bacitracin monotherapy displayed negligible activity, the addition of palmitoleic acid led to drastic potentiation of bacitracin killing similar to vancomycin (*Figures 2E and 1B–C*). Sequestration of the limited pool of bactoprenol carrier lipids by vancomycin and bacitracin reduces wall teichoic acid (WTA) biosynthesis by redirecting any remaining bactoprenol toward peptidoglycan biosynthesis (*Singh et al., 2017*). Because WTAs have been found to protect *S. aureus* from the antimicrobial activity of UFAs (*Parsons et al., 2012*), we assessed whether WTA inhibition contributed to the mechanism of synergy using the WTA inhibitor tarocin A1 (*Lee et al., 2016*). We found that inhibition of WTA biosynthesis alone does not synergize with palmitoleic acid and likely does not contribute to the potentiation of vancomycin or bacitracin (*Figure 2F*). Taken together, these results suggest that palmitoleic acid relies on antibiotic induced accumulation of bactoprenol bound intermediates.

We next validated that lipid-bound intermediates were accumulating by analyzing the membrane fraction of cell-wall precursors via high-performance liquid chromatography (HPLC). As expected, both vancomycin alone and in combination with palmitoleic acid led to the accumulation of lipid II, although dual treatment led to less accumulation of lipid II compared to vancomycin alone (*Figure 3A*). Interestingly, in the presence of palmitoleic acid, we found an unexpected peak that eluted earlier than both lipid II and lipid I (*Figure 3A–B*). Upon further inspection with a different method to improve separation of the lipid intermediate, we identified that the peak aligned with our bactoprenol standard suggesting our unknown peak represented an accumulation of bactoprenol (*Figure 3C*). Additionally, we find that even palmitoleic acid alone leads to the accumulation of bactoprenol, although a lower quantity comparative to dual treatment, indicating a novel mechanism of action for UFAs (*Figure 3A–B*).

Previous studies have determined that the accumulation of lipid II is saturated after 30 min of vancomycin treatment (*Qiao et al., 2017*). We speculate that the rapid killing seen with dual treatment in that time frame relies on maximizing lipid II and bactoprenol on the cell surface (*Qiao et al., 2017*; *Chugunov et al., 2013*). If palmitoleic acid prevents bactoprenol recycling, it is likely that the pool of lipid II available becomes significantly smaller making it easier for vancomycin to saturate its lethal target at the septum. This may explain why dual treated cells had less accumulation of lipid II than vancomycin alone (*Figure 3A*).

## Large regions of increased fluidity are generated by combining palmitoleic acid and vancomycin

Recent studies show that the bactoprenol-bound precursors, such as lipid II, alter the surrounding phospholipid bilayer (*Chugunov et al., 2013*). To accommodate the long hydrocarbon tail of these molecules, a long-lived hydrophobic and fluid microdomain are generated in the membrane, also known as regions of increased fluidity (RIFs; *Müller et al., 2016*; *Chugunov et al., 2013*). The presence

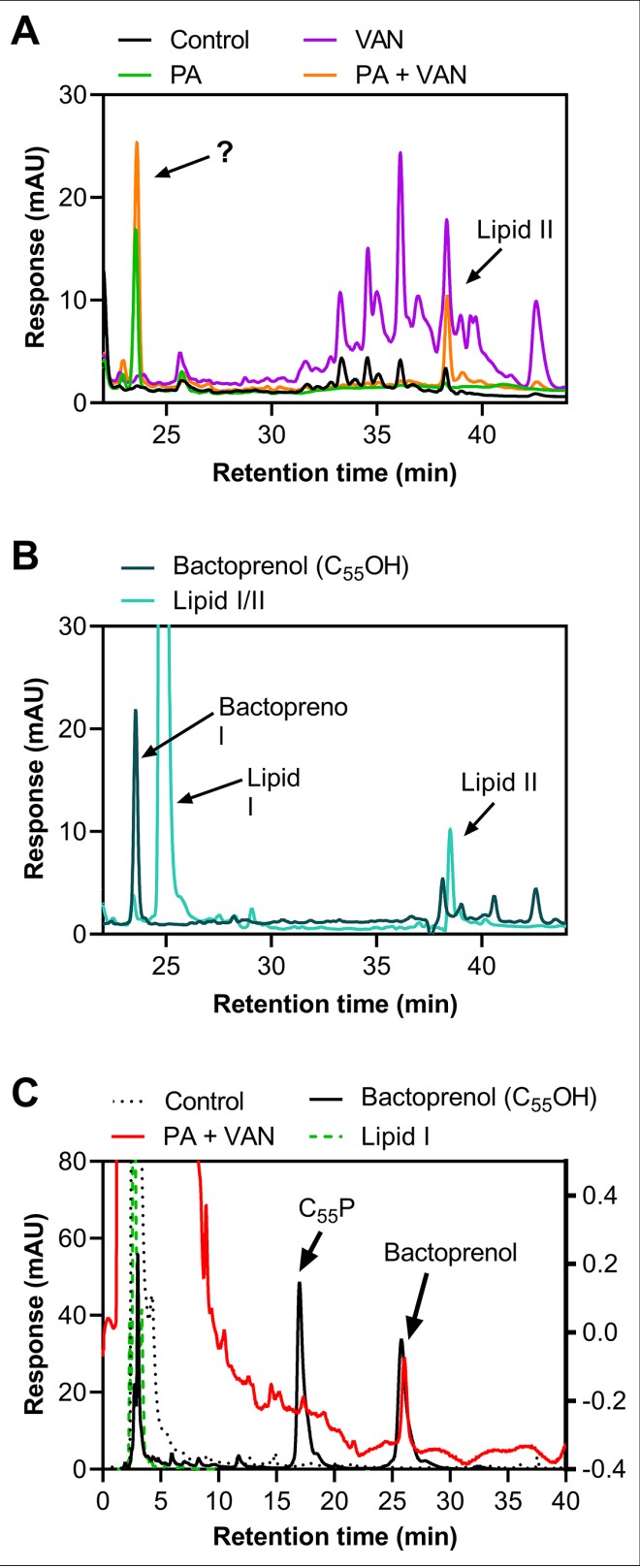

**Figure 3.** Treatment with palmitoleic acid induces accumulation of bactoprenol. (**A**) *S. aureus* was grown to exponential phase and treated with DMSO (Control), PA monotherapy (11 µg/ml), VAN monotherapy (20 µg/ml), or PA +VAN combination therapy for 30 min. The membrane fraction was extracted and run on a C4 column. The question mark indicates an unknown peak. Additional peaks in VAN monotherapy may be glycinylated versions

*Figure 3 continued on next page*

*Figure 3 continued*

of lipid II. (**B**) High-performance liquid chromatography (HPLC) of a combined lipid I and lipid II, as well as a bactoprenol standard run alongside samples in A. (**C**) Samples run in A were subsequently run with a different method to yield better separation of bactoprenol and lipid I. PA +VAN samples align with the bactoprenol standard. Right Y-axis refers to PA +VAN samples due to lower concentrations of sample after sequential use in two different columns. Chromatograms are examples of n=3 biological replicates. PA, palmitoleic acid; VAN, vancomycin.

The online version of this article includes the following source data for figure 3:

**Source data 1.** Related to *Figure 3A–C*.

---

of RIFs in the membrane also serves an essential role in the spatial organization of complex machinery that is involved in cell envelope biosynthesis. However, antibiotics that target lipid II utilize the unique characteristics of RIFs as a 'landing terrain' that distinguishes their target within a 'sea' of other lipids (*Chugunov et al., 2013*). Similar to the bulky hydrophobic nature of bactoprenol, the *cis* unsaturation in palmitoleic acid causes a 'kink' in its hydrocarbon tail which also favors a membrane environment with increased fluidity and hydrophobicity (*Mingeot-Leclercq and Décout, 2016*). Furthermore, palmitoleic acid is likely to preferentially insert at the septum where lipid II and bactoprenol are localized in RIFs. We hypothesized that cell death occurs by the induction of large RIF formation due to the combined accumulation of palmitoleic acid, bactoprenol, and lipid II, resulting in a disruption of membrane organization.

To investigate the effects of combination or monotherapy on *S. aureus* membrane fluidity and RIF organization, we utilized high-resolution fluorescence microscopy. The fluorescent lipophilic dyes nile red and DiI-C12 both insert into the membrane and preferentially accumulate and fluoresce brighter in areas of the phospholipid bilayer with increased fluidity (*Müller et al., 2016*; *Wenzel et al., 2018*; *Kucherak et al., 2010*). Unlike *Bacillus subtilis*, which is large enough to visualize RIF organization in untreated cells using DiI-C12 (*Strahl et al., 2014*), the small cell size (~0.5–1.0 μm) of *S. aureus* impedes RIF visualization (*Gray and Wenzel, 2020b*). RIFs in *S. aureus* only become clearly visible when cells are treated with membrane-active agents that cause the formation of large RIFs, as seen previously with rhodomyrtone and daptomycin (*Gray and Wenzel, 2020b*; *Saeloh et al., 2018*).

Consistent with previous studies, the untreated *S. aureus* cells exhibit smooth, consistent fluorescence across the membrane using either nile red or DiI-C12 (*Figure 4A–C*; *Saeloh et al., 2018*; *Monteiro et al., 2015*). Similarly, vancomycin monotherapy illustrated smooth membrane staining with either dye, although DiI-C12 fluorescence was brighter compared to control cells suggesting some overall increased fluidity in vancomycin treated cells (*Figure 4A–B*). Palmitoleic acid also increased overall DiI-C12 fluorescence indicating some increased fluidity in the presence of the fatty acid alone (*Figure 4A–B*). However, when cells were treated with both palmitoleic acid and vancomycin, large foci were easily observable using either nile red or DiI-C12, indicating the formation of large RIFs during dual treatment (*Figure 4A–B*). Even at sublethal concentrations of vancomycin, where no cell death occurs, RIFs are still obvious in dual treated cells, though less pronounced (*Figure 4—figure supplement 1*). Importantly, nile red staining remained uniform in cells treated with a CMAA that did not potentiate vancomycin (glycerol monolaurate) and in cells treated with palmitoleic acid combined with oxacillin (*Figure 4—figure supplement 2*). These data indicate that regardless of concentration, only cells treated with vancomycin and palmitoleic acid generate detectable RIFs.

Additionally, the RIF phenotype visualized at 10 min (*Figure 4A*) was not transient, and images taken after 30 min also displayed large foci only in dual treated conditions (*Figure 4C*). Furthermore, dual treatment resulted in nearly 50% of the cell population with at least one foci per cell at both 10 and 30 min (*Figure 4D*, *Figure 4—figure supplement 3A*). Interestingly, although there was not a difference in cell size after 10 min overall, examining the cell volume by cell cycle indicated that while dual treated cells are similar in size to control cells when in phase 3, cells in phases 1 and 2 were significantly smaller (*Figure 4—figure supplement 3B–E*). However, after 30 min, dual treated cells had a significant decrease in cell volume overall and in all cell cycle phases (*Figure 4E*, *Figure 4—figure supplement 4A–C*), suggesting that cell shrinkage occurs after prolonged exposure. We also examined the distribution of the cell cycle in each treatment group and found that at either time point, dual treated conditions had a larger proportion of the population in phase 3 (*Figure 4F*, *Figure 4—figure supplement 3F–G*). Phase 3 is a short elongation step prior to an incredibly fast separation

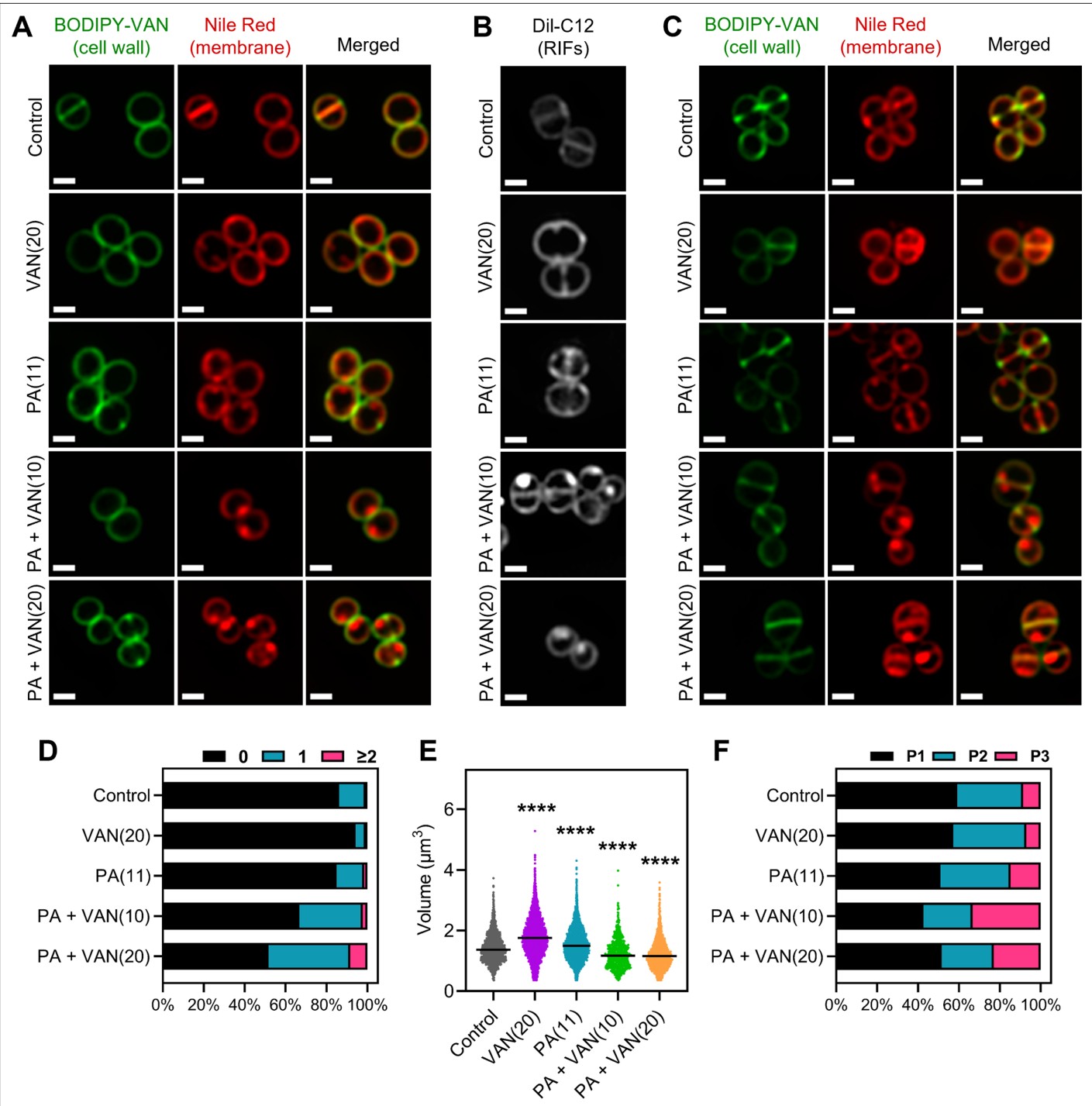

**Figure 4.** Dual treatment generates distinct fluid membrane patches. *S. aureus* (HG003) treated with DMSO, PA (11 µg/ml), VAN (20 µg/ml), PA +VAN (10 µg/ml), or PA +VAN (20 µg/ml) for (**A and B**) 10 min or (**C**) 30 min. (**A and C**) HG003 was stained with fluorescent BODIPY-labeled VAN and nile red for 5 min. (**B**) Regions of increased fluidity (RIFs) in HG003 were visualized by DiI-C12. Cells were fixed prior to imaging on an agarose pad. Images are representative of the population. Scale bar, 1 µm. (**D–F**) Bioinformatic analysis of all cells treated for 30 min and imaged, n=2 or 3 separate biological replicates. (**D**) The number of nile red foci in each cell was quantified for each treatment group and represented as a percent of the total population. (**E**) Scatter plot of cell size of each cell in the indicated treatment group. The black line represents the median. Statistical significance was determined by one-way ANOVA with Dunnett's multiple comparisons test, and all conditions had a p<0.0001, ****, compared to the control. (**F**) The cell cycle was determined for each cell in a given treatment group and illustrated as a percent of the total population; phase 1 (P1), phase 2 (P2), and phase 3 (P3).

The online version of this article includes the following source data and figure supplement(s) for figure 4:

*Figure 4 continued on next page*

*Figure 4 continued*

**Source data 1.** Related to *Figure 4D–F*.

**Figure supplement 1.** Sublethal concentrations of vancomycin with palmitoleic acid result in regions of increased fluidity.

**Figure supplement 2.** Regions of increased fluidity are not microscopically visible in the absence of synergy.

**Figure supplement 3.** Image analysis of cells treated for 10 min.

**Figure supplement 3—source data 1.** Related to *Figure 4—figure supplement 3A–F*.

**Figure supplement 4.** Dual treatment leads to a significant reduction in cell size compared to control cells regardless of cell cycle phase.

**Figure supplement 4—source data 1.** Related to *Figure 4—figure supplement 4A–C*.

**Figure supplement 5.** PA-VAN synergy is unaffected by proteins associated with flotillin microdomains.

**Figure supplement 5—source data 1.** Related to *Figure 4—figure supplement 5A–E*.

event (*Monteiro et al., 2015*; *Zhou et al., 2015*), and typically, only a small proportion of the cell population is found in phase 3, as observed in control and monotherapy treated cells (*Figure 4F*, *Figure 4—figure supplement 3F*). These results demonstrate that dual treatment leads to large RIFs, cell shrinkage, and disruption of the cell cycle.

Previous work has shown that daptomycin causes large RIF formation similar to the phenotype we see with vancomycin and palmitoleic acid (*Müller et al., 2016*). Because daptomycin can already induce RIF formation and has potent bactericidal activity alone, we did not anticipate synergy with palmitoleic acid. Indeed, when tested, palmitoleic acid did not potentiate daptomycin killing of *S. aureus* (*Figure 4—figure supplement 5A*). Additional studies have shown that in the absence of FloA or CrtM, fluid microdomains can be disturbed (*Zielińska et al., 2020*; *García-Fernández et al., 2017*). Using *S. aureus* strains from the Nebraska Transposon Mutant Library, we investigated whether *floA* or *crtM* mutants would abrogate our phenotype. Interestingly, neither disruption of *floA* nor *crtM* impacted synergy between vancomycin and palmitoleic acid and looked identical to the parental strain JE2 (*Figure 4—figure supplement 5B–D*). We further confirmed that FloA was not involved by pre-treatment with zaragozic acid, a statin that has previously been shown to disrupt FloA rafts (*García-Fernández et al., 2017*). Pre-treatment also did not diminish palmitoleic acid potentiation

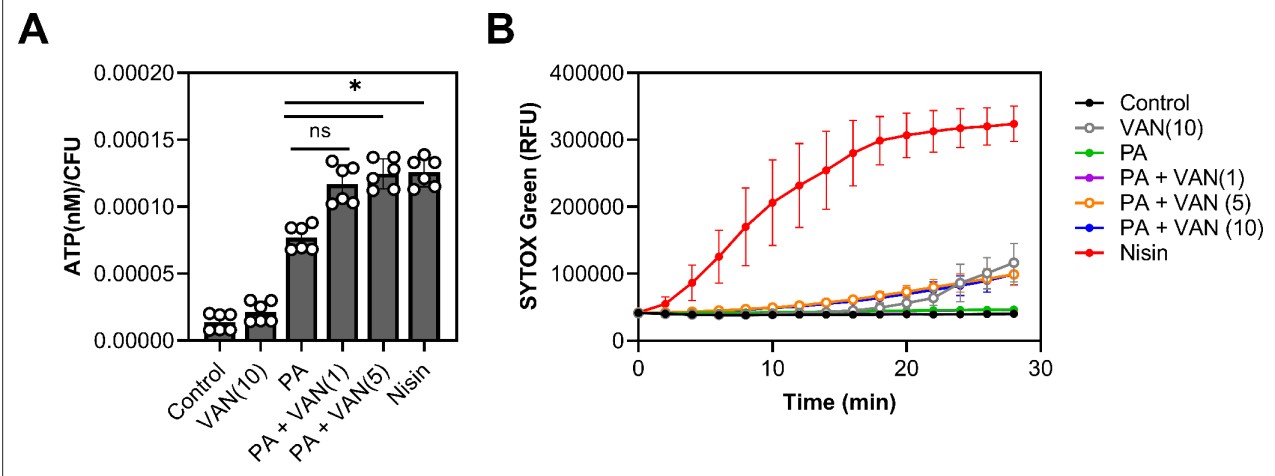

**Figure 5.** Dual treatment causes ATP leakage but not pore formation. (**A**) *S. aureus* grown to exponential phase was treated with the indicated compounds for 10 min prior to separating the supernatant from the pellet. ATP measurements were taken using a plate reader and normalized to colony forming unit (CFU). (**B**) *S. aureus* grown to exponential phase was loaded with 8 µM of SYTOX Green and aliquoted into a microtiter plate prior to treatment. Membrane permeability was measured (485/522 nm) on a plate reader every 2 min for 28 min. (**A–B**) Cells were challenged with DMSO (control), nisin (200 IU/ml), or PA (11 µg/ml)±VAN (1, 5, or 10 µg/ml) were added as indicated on the graph. Data represent the mean values from n=2 or 3 biologically independent replicates with three technical replicates each ± SD. Statistical significance was determined by one-way ANOVA with Dunnett's multiple comparisons test comparing the means of the technical replicates of each condition to PA alone. n.s. and * denote, not significant and p<0.0332, respectively.

The online version of this article includes the following source data for figure 5:

**Source data 1.** Related to *Figure 5A–B*.

of vancomycin killing (*Figure 4—figure supplement 5E*). Taken together, these data suggest that the induction of large RIFs induced by dual vancomycin-palmitoleic acid treatment is independent of factors responsible for natural, coordinated RIF formation in growing cells.

## Dual treatment increases membrane permeability without forming a pore

Palmitoleic acid has previously been reported to increase permeability, albeit at higher concentrations and at lower cell densities than examined here (*Parsons et al., 2012*; *Miller et al., 1977*; *Wang and Johnson, 1992*). However, since we observed a reduction in cell volume (*Figure 4—figure supplement 3C–E*, *Figure 4—figure supplement 4A–C*), we wanted to evaluate whether permeability is altered at sublethal concentrations of palmitoleic acid that achieve synergy with vancomycin. We first determined if combination or monotherapy led to cellular content leakage by measuring ATP in the supernatant. While the control and vancomycin monotherapy had minimal ATP leakage, all groups that were treated with palmitoleic acid had more ATP in the supernatant after 10 min. Compared to palmitoleic acid monotherapy, the dual treatment groups had significantly more ATP loss (*Figure 5A*).

We next evaluated whether leakage of cellular content was due to pore formation by using SYTOX Green which is a membrane-impermeable dye that intercalates into DNA (*Roth et al., 1997*). Even though we observed significant ATP leakage after 10 min, the membranes in all treatment groups remained impermeable to SYTOX Green after 10 min except for our positive control, nisin (*Figure 5B*). These data indicate that the outflow of cytoplasmic contents is not associated with pore formation (*Epand et al., 2016*), as is the case with nisin (*Wiedemann et al., 2004*), but instead is likely due to

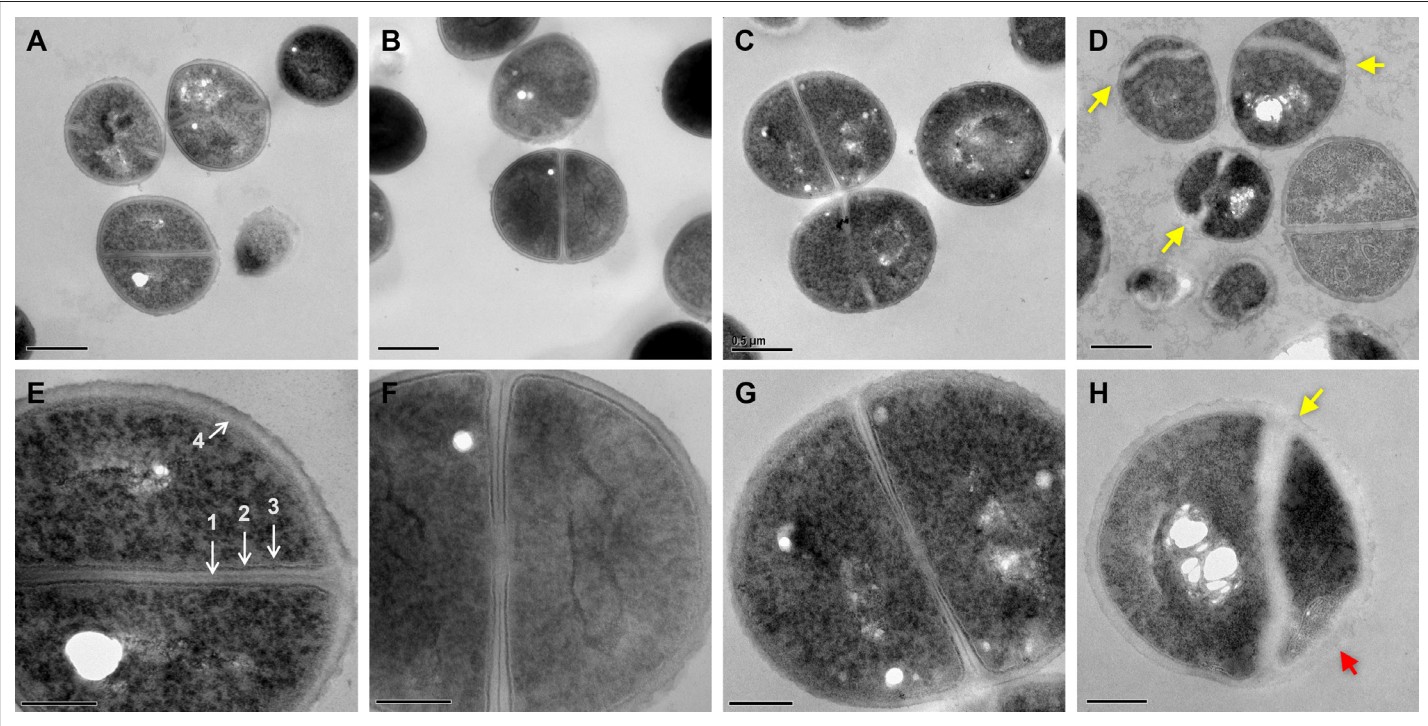

**Figure 6.** Dual treated cells display septal aberrations. The ultrastructure of *S. aureus* cells treated for 30 min was visualized by TEM. (**A–H**) *S. aureus* HG003 grown to mid-exponential phase was treated with DMSO (**A and E**), 11 μg/ml PA (**B and F**), 20 μg/ml VAN, (**C and G**) and PA +VAN (**D, H**). (**A–C and E–G**) Micrographs of cells with a cross wall at mid-cell, while (**D and H**) show cells treated with PA +VAN have deformed septa (yellow arrows) and membrane invaginations (red arrows). Magnification of 50,000× with a 0.5 μm scale bar (**A–D**) or magnification of 150,000× with a 200 nm scale bar (**E–H**). (**E**) (1) Electron-dense midline of the septum, (2) electron-dense intermediate layer, located between the (3) cell membrane and the (4) cell wall. The white holes in the cytoplasm of imaged cells are artifacts that occur during sample preparation (*Tizro et al., 1897*). PA, palmitoleic acid; VAN, vancomycin.

The online version of this article includes the following source data and figure supplement(s) for figure 6:

**Figure supplement 1.** Lipoglycopeptides do not synergize with palmitoleic acid.

**Figure supplement 1—source data 1.** Related to *Figure 6—figure supplement 1*.

increases in localized fluidity that may disturb lipid packing of the membrane and allow the escape of small molecules. This is also supported by previous work which has shown that UFAs increase membrane fluidity (*Boudjemaa et al., 2018*). However, while increased permeability may be a contributing factor, it is unlikely to be the main cause of cell death, as most CMAAs we tested (*Figure 1A*) have been shown to also increase membrane permeability, yet they did not potentiate vancomycin killing (*Radlinski et al., 2019*; *Kim et al., 2018*; *Churchward et al., 2018*; *Hess et al., 2014*).

## Dual treated *S. aureus* display aberrant septal synthesis and membrane defects

To further visualize membrane aberrations in dual-treated cells, we performed transmission electron microscopy (TEM). We observed normal cellular structure and septal formation in control, palmitoleic acid, and vancomycin monotherapy conditions (*Figure 6A–C and E–G*). In contrast, cells treated with both palmitoleic acid and vancomycin exhibited deformed septa that were misshapen and lacking an electron-dense mid-line compared to control cells (*Figure 6D and H*; yellow arrow). The dual treated cells additionally had an electron transparent region at the septa much thicker than the other conditions imaged and no discernable cell membrane or intermediate layer at the septum. Moreover, as expected based on fluorescence microscopy results, the dual treated cells had membrane aberrations that varied in size amongst the population of cells imaged (*Figure 6H*, red arrow). Interestingly, these ultrastructural alterations of the cell membrane and septa parallel those reported for lipoglycopeptide treated *S. aureus* (*Belley et al., 2009*). Palmitoleic acid did not potentiate oritavancin killing of *S. aureus*, a lipoglycopeptide, likely due to oritavancin being a potent bactericidal agent capable of inducing aberrant septal formation when used alone (*Figure 6—figure supplement 1*).

## Dual treatment delocalizes cell division and peptidoglycan biosynthesis machinery

Disruption of RIF organization by other antibiotics has been found to displace or delocalize membrane-bound proteins, such as those of cell division and peptidoglycan biosynthesis (*Müller et al., 2016*; *Saeloh et al., 2018*). The divisome of *S. aureus* is a multi-protein complex that forms a ring at the site of cell division (*Lund et al., 2018*). Cell division requires the highly coordinated efforts of peptidoglycan biosynthesis, hydrolysis, and turgor pressure (*Monteiro et al., 2015*; *Monteiro et al., 2018*). EzrA is an essential divisome protein that acts as a scaffold and regulator for the proper formation of the division ring at the septum (*Lund et al., 2018*; *Steele et al., 2011*). To determine whether the septal defects visualized with TEM are caused by an improper localization of the divisome complex, we utilized an *S. aureus* strain with a chromosomal EzrA-GFP fusion (*Saraiva et al., 2020*). We found that the control and vancomycin treated cells maintained EzrA fluorescence at the septum as a single ring at the center of the cell (*Figure 7A*). In contrast, we observed two distinct phenotypes following dual treatment with palmitoleic acid and vancomycin: delocalization of EzrA throughout the cell or multiple distinct rings within a single cell (*Figure 7A*). Interestingly, we observed a small subpopulation of cells treated with palmitoleic acid monotherapy with delocalized EzrA (*Figure 7A*). This is consistent with a previous study that suggested low concentrations of antimicrobial lipids, including palmitoleic acid, exert bacteriostatic activity as a result of momentary cell division defects prior to cell recovery (*Churchward et al., 2018*; *Cartron et al., 2014*). As controls, we evaluated whether a CMAA that did not potentiate vancomycin (glycerol monolaurate) and an antibiotic that lacked synergy with palmitoleic acid (oxacillin) maintained the septal placement of EzrA. As expected, glycerol monolaurate plus vancomycin and oxacillin plus palmitoleic acid had proper placement of cell division machinery (*Figure 7—figure supplement 1*).

Since peptidoglycan synthesis is directly linked to cell division, we next aimed to determine if dual treatment alters peptidoglycan synthesis using HADA, a fluorescent D-amino acid that is incorporated into the newly synthesized cell wall, and wheat germ agglutinin (WGA) that stains mature cell wall (*Kuru et al., 2015*). After 10 min, the lower vancomycin monotherapy concentrations had yet to inhibit cell wall biosynthesis, but the highest concentration had shut down peptidoglycan biosynthesis in a majority of the cells in any given field, as expected (*Figure 7B*, PA +VAN10). In all control or monotherapy groups, HADA was correctly located at the septa (*Figure 7B*). In contrast, when palmitoleic acid was combined with lower vancomycin concentrations that allowed HADA incorporation, we saw aberrant peptidoglycan incorporation (*Figure 7B*, gray arrows). Peptidoglycan biosynthesis is

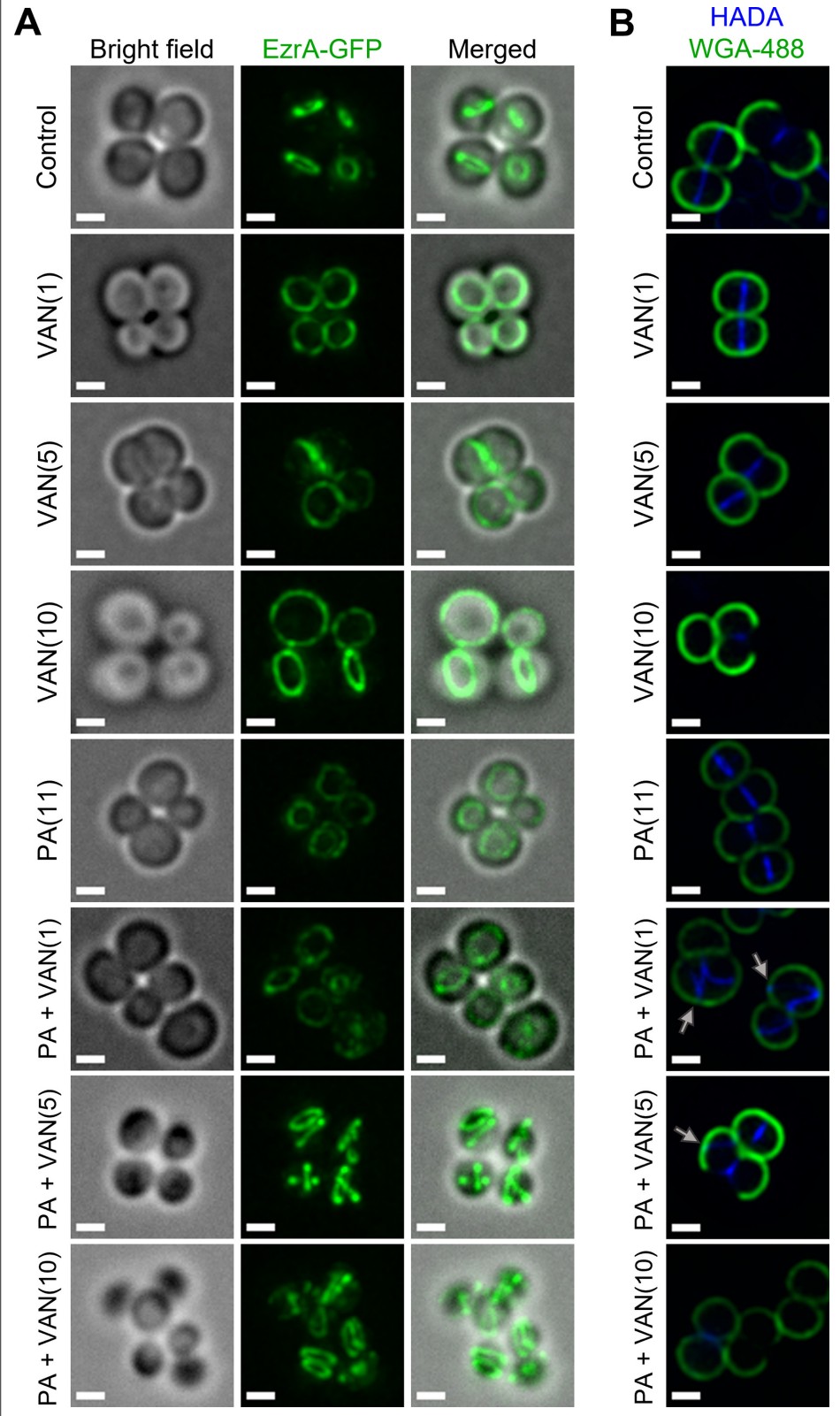

**Figure 7.** Dual treatment with vancomycin and palmitoleic acid causes septal protein delocalization. Localization of cell division and peptidoglycan biosynthesis machinery after 10 min. (**A**) HG003 with chromosomal *ezrA-gfp* was grown to exponential phase prior to treatment with DMSO (control), VAN (1, 5, or 10 µg/ml), PA (11 µg/ml), or PA +VAN at indicated concentrations in parentheses. (**B**) HG003 was grown to exponential phase and treated

*Figure 7 continued on next page*

*Figure 7 continued*

for a total of 10 min as indicated, and HADA (blue) was added after 5 min. The compounds and HADA dye were washed from the cells and stained with wheat germ agglutinin (WGA; green) for 5 min and then fixed. Gray arrows illustrated aberrant localization of peptidoglycan synthesis. (A–B) After treatment, cells were fixed prior to imaging on an agarose pad. Scale bars,1 µm. n=2 biological replicates.

The online version of this article includes the following figure supplement(s) for figure 7:

**Figure supplement 1.** Protein delocalization is absent when *S. aureus* is treated compounds that lack synergy with palmitoleic acid or vancomycin.

a highly regulated process that occurs at the site of cell division prior to rapid daughter cell separation (*Monteiro et al., 2018*). In contrast, we observed that dual treated cells began to initiate peptidoglycan synthesis at additional division planes prior to daughter cell separation further indicating cell division or cytokinesis defects (*Figure 7B*, gray arrows) (*Reichmann et al., 2019*). Together, these data suggest that vancomycin in combination with palmitoleic acid promotes divisome and peptidoglycan biosynthesis delocalization, leading to lethal aberrant septa formation.

## Palmitoleic acid sensitizes antibiotic-resistant isolates to vancomycin killing

One strategy to reduce the rise in resistance while also sensitizing already resistant and tolerant populations is to utilize chemotherapies with multiple bacterial targets (*Gray and Wenzel, 2020a*). Here, we have identified that combining palmitoleic acid and vancomycin targets multiple essential processes; thus, we next aimed to determine if dual treatment can re-sensitize resistant isolates to vancomycin. Despite the decreased susceptibility to vancomycin in characterized vancomycin-resistant *S. aureus* (VRSA) and VISA clinical isolates (with MICs of >16 µg/ml and 4 µg/ml, respectively; *Kosowska-Shick et al., 2008*), dual treatment resulted in similar killing kinetics to MSSA with 99.9% of the cells killed after only 30 min (*Figure 8A–B*). Additionally, palmitoleic acid potentiated vancomycin killing of other Gram-positive bacteria, including *E. faecalis*, *B. subtilis*, *Enterococcus faecium*, and *Staphylococcus epidermidis* (*Figure 8C–G*). Importantly, palmitoleic acid was able to potentiate vancomycin killing of vancomycin-resistant *E. faecalis* (*Figure 8D*). However, palmitoleic acid did not potentiate killing of the Gram-negative bacterial species *Escherichia coli* (*Figure 8H*), likely due to the outer membrane. These data illustrate the potency of palmitoleic acid as a vancomycin adjuvant against various Gram-positive bacteria, including those exhibiting vancomycin resistance.

## Discussion

Although new therapeutics are in the pipeline, preservation and improvement of pre-existing therapies are important for combatting rising rates of antibiotic treatment failure and antibiotic resistance. Vancomycin is a gold-standard antimicrobial as it took nearly 40 years after its introduction to the clinic before the first resistant isolate appeared (*Uttley et al., 1988*). Unfortunately, vancomycin resistance is becoming more and more prevalent, particularly amongst Enterococci, where over half of all *E. faecium* and one-third of *E. faecalis* isolates now exhibit vancomycin resistance (*De Oliveira et al., 2020*). Additionally, under certain environmental conditions including high-bacterial density, vancomycin is poorly bactericidal, facilitating antibiotic tolerance, infection recurrence, long hospital stays, and prolonged courses of parenteral antibiotics (*LaPlante and Rybak, 2004*; *Kollef, 2007*; *Liu et al., 2020*). Here, we have identified a vancomycin adjuvant, palmitoleic acid, that potentiates vancomycin killing against multiple Gram-positive bacteria. Importantly, palmitoleic acid increased the rate of killing of vancomycin at high-bacterial densities, killing over 99.9% of the bacterial populations of most isolates within 30 min, improving significant pitfalls of vancomycin monotherapy.

The research focused on the antimicrobial activity of UFAs dates back nearly a century due to their potency at high concentrations and their lack of toxicity to mammalian tissue (*Kodicek, 1945*; *Burtenshaw, 1942*; *Kabara et al., 1972*; *Lamar, 1911*). UFAs are found in nasal secretions and on the skin as an important natural protective barrier, in addition to being commonly found in our diet (*Wille and Kydonieus, 2003*; *Cartron et al., 2014*; *Neumann et al., 2015*). Mice deficient in skin monounsaturated fatty acid production show increased susceptibility to *S. aureus* in a complicated skin and soft tissue infection model (*Georgel et al., 2005*; *Takigawa et al., 2005*). Despite immense interest

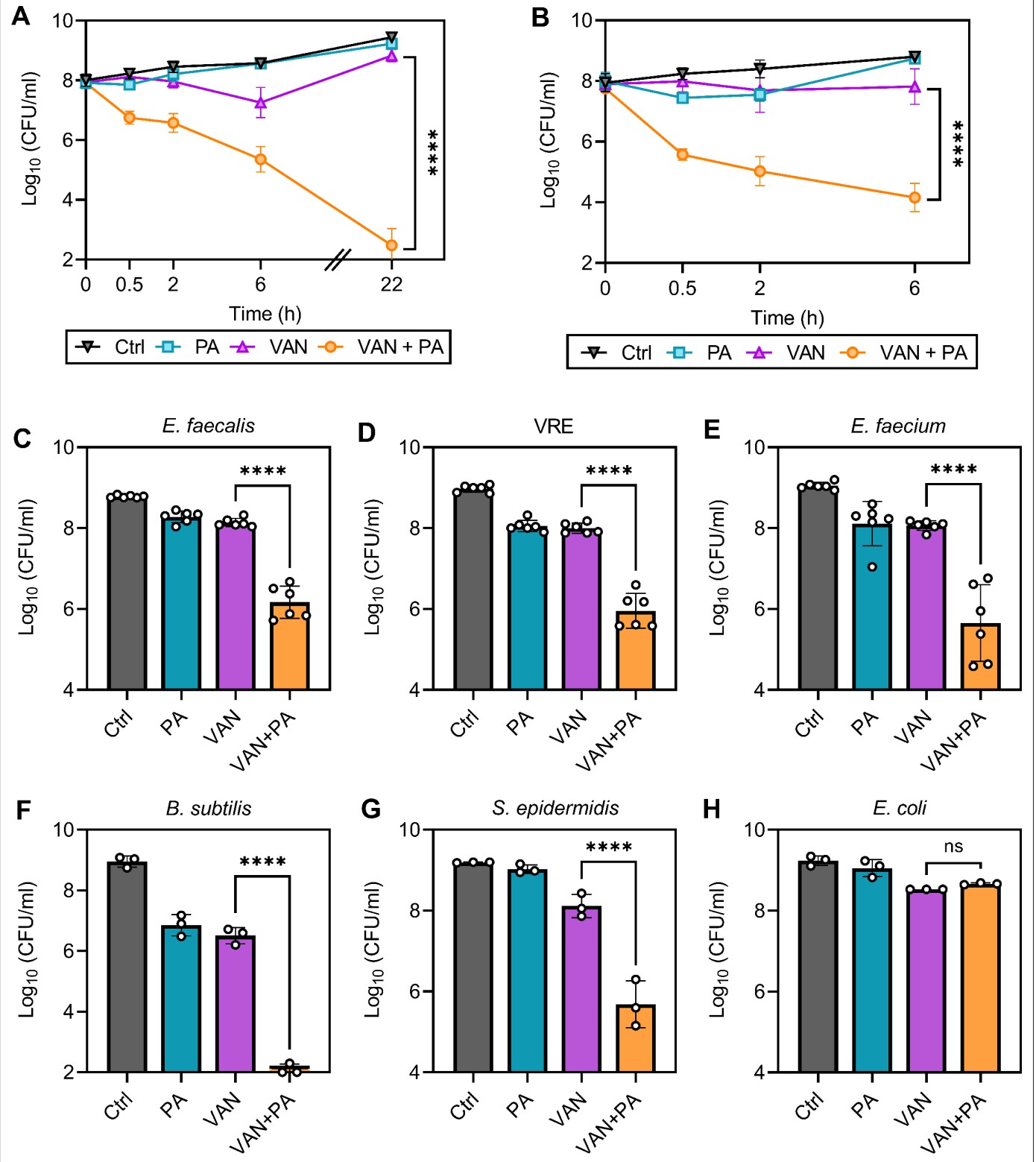

**Figure 8.** Palmitoleic acid re-sensitizes vancomycin resistant isolates. (**A**) Vancomycin-intermediate *S. aureus* (VISA) strain SA770 (***Kosowska-Shick et al., 2008***) and (**B**) vancomycin-resistant *S. aureus* (VRSA) strain VRS2 (***Kosowska-Shick et al., 2008***) were grown to exponential phase prior challenge with DMSO (Ctrl), PA (11 μg/ml), VAN (20 μg/ml), or VAN + PA. At the indicated time points, an aliquot was removed and plated to enumerate survivors. PA potentiation of other Gram-positive bacteria was determined using (**C**) *E. faecalis*, (**D**) vancomycin-resistant *E. faecalis* (VRE), (**E**) *E. faecium*, (**F**) *B. subtilis*, (**G**) *S. epidermidis*, and Gram-negative bacterial species (**H**) *E. coli*. Strains were grown to exponential phase prior to challenge with concentrations

*Figure 8 continued on next page*

*Figure 8 continued*

of PA ± VAN indicated in *Figure 8—source data 1*. After 2 hr, an aliquot was removed and plated for colony forming unit (CFU) enumeration. Data represents n=6 or 3 biologically independent replicates with error bars as SD. Significance was evaluated using Student's two-tailed unpaired *t*-test between VAN and VAN + PA conditions.

The online version of this article includes the following source data for figure 8:

**Source data 1.** Related to *Figure 8A–H*.

in antimicrobial fatty acids over the decades, a concise mechanism of action ascribed to UFAs remains unclear. Some effects attributed to UFA antibacterial activity include disruption of signal transduction, membrane depolarization, disruption in energy production, inhibition of fatty acid synthesis, increased permeability, and membrane dissolution (*Parsons et al., 2012*; *Miller et al., 1977*; *Wang and Johnson, 1992*; *Kuiack et al., 2020*; *Zheng et al., 2005*).

In this work, we have identified yet another effect elicited by UFAs, the putative accumulation of the lipid carrier, bactoprenol. Importantly, this effect was seen with sublethal concentrations of palmitoleic acid and is not a direct result of cell death. We speculate that palmitoleic acid insertion will likely localize at the septum where there is a higher concentration of fluid phospholipids that make an ideal landing terrain. The septal disruption caused by palmitoleic acid insertion likely impedes the recycling of bactoprenol and thus results in its accumulation. As palmitoleic acid does not kill bacteria at the concentrations examined here, it is likely that the cell can detoxify palmitoleic acid from the membrane and reverse bactoprenol accumulation. Moreover, individually, both vancomycin and palmitoleic acid induced accumulation of precursors were not sufficient to generate large RIFs easily detectable by microscopy. However, when applied together, the combination results in a dramatic accumulation of membrane-bound intermediates that leads to the rapid formation of large RIFs, septal disorganization, and cell death. Importantly, previous work with teixobactin has shown similar intrinsic synergy by targeting multiple lipid-bound intermediates of cell wall biosynthesis simultaneously (*Ling et al., 2015*).

Delocalization of protein machinery is likely a downstream effect of the sequestration of membrane-bound intermediates, as these intermediates are not only simple building blocks of peptidoglycan. In fact, lipid II appears to be the molecular signal for fine-tuning cell envelope biosynthesis, including during cell division (*Hardt et al., 2017*). Interfering with lipid II localization subsequently disrupts the highly coordinated cell wall biosynthesis and divisome machinery causing complex downstream effects that are irreparable (*Grein et al., 2019*; *Grein et al., 2020*). Indeed, we visualized protein delocalization in dual treated cells as defective septal formation in TEM, multiple EzrA rings, and misplaced peptidoglycan biosynthesis.

The capacity of CMAAs to sensitize vancomycin-resistant isolates is not without precedence. Interestingly, previous studies have reported that prolonged pre-treatment with antimicrobial peptides re-sensitized VRSA to vancomycin (*Mohamed et al., 2016*; *Shurko et al., 2018*). Here, we find that administration of palmitoleic acid simultaneously with vancomycin re-sensitizes resistant strains to vancomycin killing. The most common mechanism of vancomycin resistance is the maintenance of a plasmid containing the *vanA* operon. Sensing of vancomycin activates transcription of the *vanA* operon and generates an altered form of lipid II that has a terminal D-Ala-D-Lac on the stem peptide which cannot be bound by vancomycin (*McGuinness et al., 2017*). It is possible that the rapid bactericidal activity of the combination leaves little time for VRSA or VRE isolates to induce the expression of the *van* genes in response to vancomycin. Alternatively, the rapid delocalization of protein machinery seen with EzrA after only 10 min may indicate that the signal cascade required to activate the *van* operon may be disrupted by the vancomycin-palmitoleic acid combination.

Because UFAs are already a natural protectant on the skin surface and have been shown to accelerate wound healing in a rat model when applied topically (*Weimann et al., 2018*), palmitoleic acid could potentially be utilized as a topical therapeutic in conjunction with intravenous vancomycin for complicated skin and soft-tissue infections. Topical antibiotic ointments are desirable as they provide localized high concentrations without the risk of systemic toxicity. The leading over-the-counter topical ointment, Polysporin or Neosporin, contains high concentrations of bacitracin for acute skin infections (*Banerjee and Argáez, 2017*). Unfortunately, resistance to bacitracin is widespread in Staphylococci and Enterococci diminishing its utility against these infections (*Manson et al., 2004*; *Yoshida et al., 2011*). Since we find that palmitoleic acid potentiates bacitracin killing, it is possible that the addition

of palmitoleic acid to these common ointments could improve their efficacy. However additional work is needed to evaluate the potential therapeutic efficacy of palmitoleic acid as a topical ointment in vivo.

In summary, the findings presented here demonstrate a promising approach to enhancing approved therapeutics that bypasses the need to identify novel antimicrobial compounds. Focusing on antibiotic adjuvants that are naturally non-toxic to improve the efficacy of essential antibiotics can combat antimicrobial resistance and tolerance. While unraveling the mechanism of action of membrane targeting antimicrobials has consistently been challenging, we provide an important and more detailed piece of the puzzle on how targeting lipid II and the unexpected effects of palmitoleic acid on the membrane in general leads to potent synergy.

# Materials and methods

**Key resources table**

| Reagent type (species) or resource | Designation | Source or reference | Identifiers | Additional information |
|---|---|---|---|---|
| Strain and strain background (*Staphylococcus aureus*) | HG003; MSSA | doi:10.1128/IAI.00088-10 | BC1561 | |
| Strain and strain background (*S. aureus*) | LAC; CA-MRSA | doi:10.1073/pnas.0710217105 | BC1684 | |
| Strain and strain background (*S. aureus*) | JE2 wild-type | doi:10.1128/mBio.00537-12. | BC9 | |
| Strain and strain background (*S. aureus*) | JE2 *floA*::Tn | doi:10.1128/mBio.00537-12. | BC1758 | SAUSA300_1533; From NTML, strain NE184 |
| Strain and strain background (*S. aureus*) | JE2 *crtM*::Tn | doi:10.1128/mBio.00537-12. | BC440 | SAUSA300_2499; From NTML, strain NE1444 |
| Strain and strain background (*S. aureus*) | HG003 *ezrA-gfp* | doi:10.1038/nature25506 | BC1552 | *ezrA-gfp* was transduced from ColpSGEzrA-GFP into HG003 background for this work |
| Strain and strain background (*S. aureus*) | VISA | doi:10.1128/AAC.01073-08 | SA770; BC1477 | |
| Strain and strain background (*S. aureus*) | VRSA | doi:10.1128/AAC.01073-08 | VRS2; BC1479 | Vancomycin-resistant |
| Strain and strain background (*Enterococcus faecalis*) | VRE | doi:10.1128/iai.00425- 15 | V583; BC231 | Vancomycin-resistant |
| Strain and strain background (*E. faecalis*) | OG1 | ATCC | BC524 | ATCC 47077 |
| Strain and strain background (*Enterococcus faecium*) | Clinical isolate | doi:10.1038/s41564-021-00966-0 | BC1540 | |
| Strain and strain background (*Bacillus subtilis*) | BS49 | doi:10.1371/journal.pgen.1006701 | BC209 | |
| Strain and strain background (*Escherichia coli*) | MG1655 | ATCC | BC15 | ATCC 47076 |
| Strain and strain background (*Staphylococcus epidermidis*) | CSF41498 | doi:10.1128/jb.0194 6-14 | BC17 | |
| Chemical compound and drug | Rhamnolipids | AGAE Technologies | Cat#R90 | |

*Continued on next page*

*Continued*

| Reagent type (species) or resource | Designation | Source or reference | Identifiers | Additional information |
|---|---|---|---|---|
| Chemical compound and drug | Palmitoleic acid | Cayman Chemical | Cat. #10009871 | |
| Chemical compound and drug | Linoleic acid | Cayman Chemical | Cat. #90150 | |
| Chemical compound and drug | Glycerol monolaurate | Cayman Chemical | Cat.#28170 | |
| Chemical compound and drug | Lauric acid | Cayman Chemical | Cat.#10006626 | |
| Chemical compound and drug | Adarotene | Sigma-Aldrich | Cat.#SML2061 | |
| Chemical compound and drug | Benzyl alcohol | ThermoFisher | Cat.#A396500 | |
| Chemical compound and drug | Vancomycin | ThermoFisher | Cat.#AAJ6279003 | |
| Chemical compound and drug | Fosfomycin | Sigma-Aldrich | Cat.#P5396 | |
| Chemical compound and drug | Oxacillin | ThermoFisher | Cat.#AC45544005 | |
| Chemical compound and drug | Nafcillin | ThermoFisher | Cat.#AC46138001 | |
| Chemical compound and drug | Bacitracin | ThermoFisher | Cat.#BP29501 | |
| Chemical compound and drug | Tarocin A1 | Sigma-Aldrich | Cat.#SML1677 | |
| Chemical compound and drug | Daptomycin | MedChem Express | Cat.#HY-B0108 | |
| Chemical compound and drug | zaragozic acid | Sigma-Aldrich | Cat.#z2626 | |
| Chemical compound and drug | Oritavancin | Cayman Chemical | Cat.#24091 | |
| Chemical compound and drug | AsKO | Sigma-Aldrich | Cat.#A6756 | |
| Chemical compound and drug | CCCP | Cayman Chemical | Cat.#25458 | |
| Other | BODIPY-VAN | Invitrogen | Cat.#V34850 | Fluorescently labeled vancomycin, labeling cell wall, *Figure 4* |
| Other | Nile red | Sigma-Aldrich | Cat.#72485 | Cell membrane dye, *Figure 4* |
| Other | DiI-C12 | ThermoFisher | Cat.#D383 | RIF specific dye, *Figure 4* |
| Other | HADA | ThermoFisher | Cat.#66475 | Dye that stains new cell wall, *Figure 7* |
| Other | WGA-488 | ThermoFisher | Cat.#W11261 | Dye that stains mature cell wall, *Figure 7* |
| Other | Sytox green | ThermoFisher | Cat.#S7020 | Cell viability dye, *Figure 5* |
| Commercial assay or kit | BacTiter-Glo Microbial Viability Kit | Promega | Cat.#G8232 | |
| Software and algorithm | Graphpad Prism | Graphpad.com | Prism 9 | |

*Continued on next page*

*Continued*

| Reagent type (species) or resource | Designation | Source or reference | Identifiers | Additional information |
|---|---|---|---|---|
| Software and algorithm | Biorender | Biorender.com | | |
| Software and algorithm | Metamorph 7.10 | Molecular Devices | | Acquisition software |
| Software and algorithm | Autoquant | Media cybernetics | Version 3.1.3 | Deconvolution software |
| Software and algorithm | FIJI | ImageJ | v1.53q | |

## Bacterial growth and conditions

*S. aureus* strains HG003 (MSSA; *Herbert et al., 2010*), HG003 *ezrA-gfp*, LAC (CA-MRSA; *Kennedy et al., 2008*), JE2, JE2 *floA::tn*, JE2 *crtM::tn* (*Fey et al., 2013Fey et al., 2013*), SA770 (VISA), and VRS2 (VRSA; *Kosowska-Shick et al., 2008*); *B. subtilis* strain BS49 (*Anjuwon-Foster and Tamayo, 2017*), *E. coli* strain MG1655 (*Blattner et al., 1997*), and *S. epidermidis* strain CSF41498 (*Conlon et al., 2014*) were routinely cultured in tryptic soy broth (TSB, Remel) at 37°C with 225 rpm shaking. *E. faecalis* strains OG1 and V583 (*Maharshak et al., 2015*) and a clinical *E. faecium* isolate (*Aggarwala et al., 2021*) were cultured in Brain Heart Infusion Broth (Oxoid) at 37°C with shaking (*E. faecium*) or statically (*E. faecalis* strains). The EzrA-GFP was transduced from the chromosome of ColpSGEzrA-GFP (*Monteiro et al., 2018*) to HG003 by phage transduction using phage 80α as previously described (*Schneewind and Missiakas, 2014*).

## In vitro antibiotic survival assays

Antibiotic survival assays were performed as previously described (*Sidders et al., 2021*). Briefly, overnight cultures were diluted 1:1000 into fresh media and grown for 3–5 hr depending on the strain to yield a starting inoculum in late exponential phase (~$2 \times 10^8$ CFU/ml). The time required after back diluting an overnight culture to reach the starting inoculum was determined by performing growth curves of each strain (data not shown). Concentrations of antibiotics used in survival assays can be found in the figure legends or source data. At the indicated time points, an aliquot of cells was washed twice with PBS, serially diluted, and plated on tryptic soy agar to enumerate CFU. For assays with CCCP, AsKO, and zaragozic acid, these compounds were added 30 min prior to antibiotic challenge. Statistical analysis is indicated in the figure legends.

## Membrane integrity assays

Membrane permeability was measured using the SYTOX Green (Invitrogen) as previously described (*Yasir et al., 2019*). Briefly, measurements were carried out in a Synergy H1 (BioTek) plate reader equipped with 480 nm excitation and 522 nm emission filters. *S. aureus* HG003 was grown in TSB as described above. After reaching the desired starting CFU, cells were washed thrice with PBS + 10% TSB and resuspended. Cells were loaded with SYTOX Green for 15 min at room temperature in the dark. Subsequently, 100 μl of cells were added to each well of a black flat bottom 96-well polystyrene plate that was non-binding (Corning, 3991) to prevent the necessity of BSA being added. The final concentration of SYTOX Green was 8 μM. Compounds were added at concentrations indicated in the figure legend, and reads were taken every 2 min for 28 min while shaking.

ATP levels were evaluated using the BactiterGLO kit (Promega) per manufacturer's specifications and normalized to CFU that was taken after 10 min of treatment.

## Accumulation of cell wall precursors

*S. aureus* HG003 was grown to ~$1 \times 10^8$ CFU/mL before being treated with vancomycin (10 μg/mL), palmitoleic acid (11 μg/mL), or the combination of both for 30 min. Cell numbers were documented after treatment, and cells were harvested at 8000 rpm for 10 min before being resuspended in 1× PBS. Lipids were extracted by addition of an equal volume of *n*-butanol-pyridine acetate (2:1 [v/v] pH 4.2), followed by vigorous vortexing and 3 min of centrifugation at 13,500 rpm using a microfuge to achieve phase separation, as previously described (*Müller et al., 2012*). The lipid containing upper phase was collected, and extraction was repeated with the lower phase. Upper phases were combined, washed twice with cold water (pH 4.2), and fully dried by desiccation. Dried lipids were resuspended in *n*-butanol with adjustment of the resuspension volume according to the normalized cell number. Analysis

of lipid content was achieved by thin layer chromatography developed in the solvent according to Rick (CHCl₃:MeOH:H₂O:NH₄OH [88:48:10:1]) followed by PMA staining as described before (*Rick et al., 1998*; *Schneider et al., 2004*). Additionally, samples were subjected to HPLC.

## HPLC of cell wall precursors

HPLC was performed on an Agilent 1260 Infinity II System (Agilent Technologies, Santa Clara, CA, USA) equipped with a VWD detector. Separation of lipid II was achieved on a MultoHigh Bio 300 – C4 (4 × 125 mm, 5 µm) with a flow rate of 0.5 ml/min at 30°C, using a linear gradient of 3:1:1 ($H_2O$:MeOH:IPA) for 3 min, which increased to 100% 0.5:1:1 ($H_2O$:MeOH:IPA) in 35 min and was maintained for 5 min. Before the following run, the gradient was returned to 0% in 2 min and was retained for 5 min. Both solvent mixtures were buffered with 10 mM $H_3PO_4$.

Separation of bactoprenol and lipid I was achieved on an Agilent Poroshell 120 C18 column (3 × 150 mm, 2.7 µm) with a flow rate of 0.3 ml/min at 30°C using an isocratic gradient for 50 min of 4:1 (MeOH:IPA) buffered with 10 mM H3PO4.

The data was analyzed by comparing the peaks against standards of different purified cell wall precursors. Data analysis was performed using OpenLab CDS software version 2.6.

## Fluorescence microscopy

All imaging was performed using *S. aureus* HG003 grown as described above prior to challenge with compounds at concentrations and time points indicated in figure legends.

To evaluate morphological changes, 1 ml of culture was incubated for 5 min at 37°C with 5 µg/ml of nile red (Sigma) to stain the membrane, and the cell wall was labeled with 1 µg/ml of an equal mixture of vancomycin (Alfa Aesar) and a BODIPY FL conjugate of vancomycin (Invitrogen). Excess dye was removed by washing with TSB. To determine if membrane perturbations were RIFs, the lipophilic dye DiI-C12 (Invitrogen) was used as described (*Wenzel et al., 2018*). Briefly, overnight cultures were diluted 1:1000 into fresh TSB containing 1% DMSO and 1 µg/ml DiI-C12. Cells were grown to ~2 × 10⁸ CFU/ml and washed four times with pre-warmed TSB containing 1% DMSO. Cells were then treated for 10 min and washed to remove excess antibiotic and dye. Washes and incubation steps were performed in a warm room to maintain samples at 37°C for optimal DiI-C12 solubility. For visualization of divisome proteins, overnight cultures of HG003 *ezrA-gfp* were diluted 1:1000 into fresh TSB and grown to ~2 × 10⁸ CFU/ml. Cells were then treated for 10 min prior to washing with TSB. To visualize peptidoglycan biosynthesis, 1 ml of culture was incubated with the fluorescent D-amino acid HADA (Tocris Bioscience) at a final concentration of 250 µM for 5 min. The cells were subsequently washed with TSB and incubated with WGA tagged with Alexa Fluor 488 (WGA-488, Invitrogen) at a final concentration of 2 µg/ml for 5 min; excess WGA-448 was removed by washing with TSB. Both incubation steps were carried out at 37°C with agitation.

After staining, *S. aureus* was fixed with 4% paraformaldehyde made fresh day of for 20 min at room temperature with agitation. Cells were then washed with PBS and resuspended in 1st/10th the starting volume. Samples were imaged within 24 hr of fixation by pipetting 2 µl of sample onto an agarose pad (made as previously described *Skinner et al., 2013*) and mounted onto a #1.5 cover slip.

Image Z-stacks were acquired using an Olympus IX81 inverted microscope fitted with a Hamamatsu ORCA-Flash4.0 V3 camera, X-Cite XYLIS XT720L (385) illumination, 100×/1.4 Oil UPlan S Apo PSF objective lens, and Metamorph 7.10 acquisition software (Molecular Devices). Z spacing was 0.2 µm, and pixel size in XY was 0.064 µm. Our Z-stacks ranged in thickness from 2.6 to 4.0 µm. We acquired images for all channels at each Z position before moving to the next Z position. Nile red was imaged with 560/25 excitation and 607/36 emission filters. BODIPY-VAN, EzrA, and WGA-488 were imaged with 485/20 excitation and 525/30 emission filters. HADA was imaged with 387/11 excitation and 440/40 emission filters. DiI-C12 was imaged with 572/35 excitation and 632/60 emission filters. All images were acquired with the same exposure settings. No pixels had saturated intensities. Z-stacks were subsequently deconvolved using Autoquant software version 3.1.3 with default settings. All images in a figure were processed in FIJI and have the same display adjustments. A single Z-plane that was determined to be the middle cross-section of *S. aureus* cells in the field of view was used for representative images of nile red, VAN-FL, WGA-488, and DiI-C12. A max projection of all Z-stacks was used for the representative images of HADA and EzrA-GFP.

## Image analysis

Image analysis was performed using Python with scikit-image (0.18.2) image processing library (*van der Walt et al., 2014*) and Fiji (v1.53q; *Schindelin et al., 2012*). The full analysis code is available on Github (copy archived at *Sidders et al., 2023*), while all raw imaging data and key processed steps were deposited in Zenodo repository.

For quantification of bacterial cell features (volume, phase, and foci number), a middle cross-section from confocal z-stacks was used. This plane was selected automatically by finding the frame with maximum cell area from thresholded (Otsu) images of the cell wall (BODIPY-VAN). Detection of cells was performed in the membrane channel (Nile Red) using cellpose segmentation algorithm (*Stringer et al., 2021*). Total number of detected cells (all conditions) >32 k. Volume of cells was calculated based on the major and minor axis length of segmented cells as described previously (*Monteiro et al., 2015*). Membrane foci were segmented using Maximum Entropy Threshold (Fiji). Phases of the cell cycle were determined using a classification network trained using fastai deep learning library (2.5.3; *Howard and Gugger, 2020*). Briefly, a Resnet50 pre-trained network (fastai) was trained using manually annotated sets of cells (phase 1–150, phase 2–150, phase 3–150, and missegmented cells - 80) achieving error rate <4%. Trained classifier was used to predict the cell cycle phase of remaining cells.

## Transmission electron microscopy

*S. aureus* was grown and challenged as described above. Cultures were centrifuged at $1500 \times g$, and the supernatant was removed. Bacterial cells pellets were resuspended in 4% paraformaldehyde/2.5% glutaraldehyde in 0.15 M sodium phosphate buffer, pH 7.4, for 1 hr at room temperature and stored at 4°C. Following three washes with 0.15 M sodium phosphate buffer, pH 7.4, the fixed cell pellets were post-fixed in 1% osmium tetroxide in 0.15 M sodium phosphate buffer, pH 7.4 for 45 min and washed three times with deionized water. The samples were gradually dehydrated with ethanol (30, 50, 75, 100, and 100%) and propylene oxide. The cell pellets were infiltrated with a 1:1 mixture of propylene oxide for and Polybed 812 epoxy resin overnight, followed by a 1:2 mixture of propylene oxide for and Polybed 812 epoxy resin for 3 hr and transferred to 100% resin to embed overnight (Polysciences, Inc,Warrington, PA, USA). Ultrathin sections (70–80 nm) were cut with a diamond knife and mounted on 200 mesh copper grids followed by staining with 4% aqueous uranyl acetate for 12 min and Reynold's lead citrate for 8 min (*Reynolds, 1963*). Samples were observed using a JEOL JEM-1230 TEM operating at 80 kV (JEOL USA, Inc, Peabody, MA, USA), and images were acquired with a Gatan Orius SC1000 CCD Digital Camera and Gatan Microscopy Suite 3.0 software (Gatan, Inc, Pleasanton, CA, USA).

## Acknowledgements

This work was supported in part by NIH grants R01AI137273, Burroughs Wellcome Fund, and Cystic Fibrosis Foundation Research Grant to BPC. Additionally, this work was supported by the Deutsche Forschungsgemeinschaft project-ID 398967434 – TRR261 to TS. We thank Marianna Pinho for providing COL-*ezra-gfp,* Kim Lewis for providing VISA and VRSA isolates, and Jeremiah Faith for providing the clinical *E faecium* isolate. We thank Pablo Ariel at the Microscopy Service Laboratory (MSL) at UNC for helping optimize fluorescence microscopy experiments. We thank Kristen White at the MSL for performing TEM imaging and processing. The MSL, Department of Pathology and Laboratory Medicine, is supported in part by P30 CA016086 Cancer Center Core Support Grant to the UNC Lineberger Comprehensive Cancer Center. The image analysis was performed at the Bioinformatics and Analytics Research Collaborative (BARC) which is supported in part by the UNC School of Medicine Strategic Plan and Office of Research. KMK is supported by grant number 2020–225716 from the Chan Zuckerberg Initiative DAF, an advised fund of Silicon Valley Community Foundation. We thank Virginie Papadopoulou, Phil Durham, Lauren Radlinski, Baggio Evangelista, and Joey Ragusa for thoughtful discussions.

# Additional information

## Competing interests

Sarah E Rowe, Brian P Conlon: is co-inventor on a provisional patent (WO2019018594A1) describing the use of membrane acting agents for potentiating antibiotic efficacy. The other authors declare that no competing interests exist.

## Funding

| Funder | Grant reference number | Author |
| --- | --- | --- |
| National Institutes of Health | R01AI137273 | Brian P Conlon |
| Burroughs Wellcome Fund | | Brian P Conlon |
| Cystic Fibrosis Foundation | | Brian P Conlon |
| Deutsche Forschungsgemeinschaft | 398967434 – TRR261 | Tanja Schneider |
| Silicon Valley Community Foundation | 2020–225716 | Katarzyna M Kedziora |

The funders had no role in study design, data collection and interpretation, or the decision to submit the work for publication.

## Author contributions

Ashelyn E Sidders, Conceptualization, Data curation, Formal analysis, Validation, Investigation, Visualization, Methodology, Writing - original draft, Project administration, Writing – review and editing; Katarzyna M Kedziora, Resources, Data curation, Software, Formal analysis, Funding acquisition, Validation, Methodology, Writing – review and editing; Melina Arts, Jan-Martin Daniel, Stefania de Benedetti, Resources, Investigation, Visualization, Writing – review and editing; Jenna E Beam, Duyen T Bui, Investigation, Writing – review and editing; Joshua B Parsons, Supervision, Writing – review and editing; Tanja Schneider, Resources, Funding acquisition, Project administration; Sarah E Rowe, Supervision, Investigation, Writing – review and editing; Brian P Conlon, Conceptualization, Resources, Supervision, Funding acquisition, Project administration, Writing – review and editing

## Author ORCIDs

Ashelyn E Sidders (ID) http://orcid.org/0000-0002-8700-261X
Tanja Schneider (ID) http://orcid.org/0000-0001-7269-4716
Sarah E Rowe (ID) http://orcid.org/0000-0001-8955-359X
Brian P Conlon (ID) http://orcid.org/0000-0002-2155-8375

## Decision letter and Author response

Decision letter https://doi.org/10.7554/eLife.80246.sa1
Author response https://doi.org/10.7554/eLife.80246.sa2

# Additional files

## Supplementary files

- MDAR checklist

## Data availability

The raw numerical data for figures 1, 2, 3, 5, and 8 are provided in Source Data files for each figure. The full analysis code for Figure 4 is available on Github (copy archived at *Sidders et al., 2023*) while all raw imaging data and key processed steps were deposited in Zenodo repository.

The following dataset was generated:

| Author(s) | Year | Dataset title | Dataset URL | Database and Identifier |
|-----------|------|---------------|-------------|-------------------------|
| Sidders AE, Kedziora KM, Arts M, Daniel JM, de Benedetti S, Beam JE, Bui DT, Parsons JB, Schneider T, Rowe SE, Conlon BP | 2023 | Antibiotic-induced accumulation of lipid II synergizes with antimicrobial fatty acids to eradicate bacterial populations - confocal dataset | https://doi.org/10.5281/zenodo.7753640 | Zenodo, 10.5281/zenodo.7753640 |

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
