## [Editor Report]

The authors present how unsaturated fatty acids modulate the bactericidal effect of the antibiotic vancomycin. The authors find that palmitoleic acid significantly increases the bactericidal activity of vancomycin and reveal the mechanism responsible. The key findings will be of interest to a broad audience of researchers focused on microbiology, host-pathogen interactions, and antimicrobial development, as well as to clinicians that treat antibiotic-recalcitrant infections.

---

## [Decision Letter]

**Decision letter after peer review:**

Thank you for submitting your article "Antibiotic-induced accumulation of lipid II synergizes with antimicrobial fatty acids to eradicate bacterial populations" for consideration by *eLife*. Your article has been reviewed by 3 peer reviewers, and the evaluation has been overseen by a Reviewing Editor and Wendy Garrett as the Senior Editor. The following individual involved in the review of your submission has agreed to reveal their identity: Jim Cassat (Reviewer #1).

The reviewers have discussed their reviews with one another, and the Reviewing Editor has drafted this to help you prepare a revised submission. Please attend to the points listed by the editor, the full reviews are appended below for your information only.

Essential revisions:

1. It is unclear if bacteria in other growth phases would be killed by dual PA and Vanc therapy. Please test if bacterial cultures approaching the stationary phase are also effectively killed by UFAs plus Vanc. Results will dictate how much can be concluded about tolerant cells.

2. The proposed mechanism of UFA/Vanc synergy of accumulation of lipid II and alterations of membrane fluidity is partially demonstrated. Please strengthen these findings with the following experiments:

– Assess lipid II buildup under treatment conditions employed and thus determine if there is a correlation between the degree of lipid II buildup and the extent of killing. Does the extent of lipid II buildup change over time under vancomycin/bacitracin treatment and if so, does the magnitude and/or rapidity of PA bactericidal effects vary depending on levels of accumulated lipid II at the time of PA addition?

– Previous work has shown disruption/delocalisation of the divisome via treatment with daptomycin (Muller et al., 2016 PNAS) or zaragozic acid (Garcia-Fernandez et al., 2017 Cell). A floA mutant also has disrupted fluid domains and a crt mutant has a partial phenotype. Overexpression of GpsB antisense mRNA from a multicopy plasmid leads to membrane patches very similar to what is observed here (Eswara et al. *eLife* 38856 2018 Figures 2 and 4). Based on the proposed mechanism here, test whether the effect of the PA treatment is abrogated in some of these conditions.

– Figure 6 does not show equivalent experiments with fatty acids that do not synergise with vancomycin. Please provide this control.

– Figure 4 shows that PA causes membrane fluid patches, which may or may not contribute to enhanced killing. Please test whether non-potentiating fatty acids cause patches. If PA specifically causes patches, it would provide stronger support of the model according to which these fluid domains contribute to killing.

– Lines 280-282: 'Interestingly, these ultrastructural alterations of the cell membrane and septa parallel those reported for lipoglycopeptide treated *S. aureus*', implies that the combination of vancomycin and PA mimics lipoglycopeptide. Please test by combining a lipoglycopeptide with PA, which would presumably not synergise.

3. Figure 3b would benefit from a positive control to clarify the biological significance of the differences seen. What is the ATP loss with a membrane-damaging agent such as nisin?

4. Lines 205 and 206 state: 'Vancomycin induces palmitoleic acid incorporation at the septum leading to disruption of functionalized fluid regions'. However, the fluid patches shown in figure 4 are not associated with septa. For example, in Figure 4b, Van +PA(10), none of the patches appear to be near the septa. Please rephrase.

5. Figure 1b, c. A t-test is inappropriate here because of the multiple comparisons (it is stated that no significant differences were detected unless indicated). Please use a 2-way repeated measures ANOVA. Figure 3b. Were stats done on 9 data points for each condition (3 independent replicates X 3 technical repeats?) as indicated by the figure? This would infer a much greater statistical power than is appropriate and increase the likelihood of P<0.05. Statistical testing should be done with the means of the 3 independent experiments.

6. Lines 245-247 state: '…dual treated cells had a significant decrease in cell volume after 30 minutes (Figure 4F), suggesting that after prolonged exposure dual treated cells shrink, likely reflecting the leakiness of the membrane'. This set of experiments needs strengthening. Measure cells in each of the growth phases and determine whether this affects cell size. It appears that VAN-treated, and possibly PA-treated, cells have a greater volume than control. This would not fit with the conclusions in the text or figure legend stating that all conductions had p<0.0001 relative to control.

*Reviewer #1 (Recommendations for the authors):*

The authors should report the growth phase of bacteria tested in all in vitro experiments. The methods state that overnight cultures were back-diluted at 1:1000, but it is unclear how long these back-diluted cultures were grown. This is important because although persisters are likely present at the high inoculum chosen, other mechanisms of tolerance are more appropriately modeled using bacteria that reach the stationary phase. If the experiments were conducted with exponential phase bacteria, it would improve the manuscript and the interpretation of the efficacy of dual therapy to conduct new experiments with stationary phase bacteria.

*Reviewer #2 (Recommendations for the authors):*

The manuscript by Sidders et al., presents the interesting finding that palmitic acid selectively potentiates the vancomycin-mediated killing of *S. aureus*. This is an Interesting observation and the authors attempt to elucidate the underlying mechanism. However, the work lacks some important controls and requires better statistical analysis.

Specific points:

1. Figure 1b, c. A t-test is inappropriate here as you have to account for multiple comparisons, even if you don't make them (the more things you measure, the greater the likelihood of getting spurious significance and so you need to account for multiple data points). You should use a 2-way repeated measures ANOVA.

2. Figure 1. The legend is unclear. For b and c, the legend states the test was done at 6h, but it's clearly done at 22 h for panel c. Further, the legend states no significant difference unless indicated but it's not clear if that means for panel a or all graphs? For b and c, I would suggest indicating points that significantly differ from control.

3. The legend for figure 3 states: 'Dual treatment impacts membrane permeability but not energetics' but there are no statistical tests shown to support this for membrane permeability and the text states '…some depolarization was observed, however, this was negligible when compared to gramicidin, the pore-forming positive control…' (lines 182-183). Therefore, it's not clear whether the authors believe that significant impacts on the membrane are occurring.

4. Figure 3b. Were stats done on 9 data points for each condition (3 independent replicates X 3 technical repeats?) as indicated by the figure? This would infer a much greater statistical power than is appropriate and increase the likelihood of P<0.05. Statistical testing should be done with the means of the 3 independent experiments.

5. Figure 3b would benefit from a positive control to clarify the biological significance of the differences seen. What is the ATP loss with a membrane-damaging agent such as nisin?

6. Lines 205 and 206 state: 'Vancomycin induces palmitoleic acid incorporation at the septum leading to disruption of functionalized fluid regions'. However, the fluid patches shown in figure 4 are not associated with septa. For example, in Figure 4b, Van +PA(10), none of the patches appear to be near the septa. Furthermore, there is no evidence that functional domains are disrupted at this point – evidence comes later in the manuscript via mis-localisation of cell wall machinery as observed using gfp-tagged enzyme (but see point 12).

7. Figure 4 shows that PA causes membrane fluid patches, which may or may not contribute to enhanced killing. However, there is no examination of whether non-potentiating fatty acids cause patches. If it can be shown that PA specifically causes patches then that would provide some evidence that these fluid domains contribute to killing. This would seem to be a critically important experiment.

8. Figure 4d: How many independent experiments were done here? Please show SD for data. Were any statistical tests done?

9. Lines 245-247 state: '…dual treated cells had a significant decrease in cell volume after 30 minutes (Figure 4F), suggesting that after prolonged exposure dual treated cells shrink, likely reflecting the leakiness of the membrane'. This statement lacks any supporting evidence/citations and it's much more likely that any effects on size are governed by the cell wall rather than the membrane since this is the outer-most structure, perhaps reflective of the observed shift in the proportion of cells in the growth phase 3. A useful control here would be to measure cells in each of the growth phases and determine whether this affects cell size.

10. Lines 280-282: 'Interestingly, these ultrastructural alterations of the cell membrane and septa parallel those reported for lipoglycopeptide treated *S. aureus*', presumably suggesting that the combination of vancomycin and PA mimic lipoglycopeptide. This could be tested by combining a lipoglycopeptide with PA, which would presumably not synergise.

11. Figure 6 does not show equivalent experiments with fatty acids that do not synergise with vancomycin. This is another critical experiment if mis-localisation of cell wall machinery is to be considered to contribute to the phenotype.

12. The susceptibility of strains to vancomycin is unclear. How much less susceptible are the VISA/VRSA isolates relative to the strain used initially? MIC values should be provided.

13. It's not clear what the dose-response profile for PA looks like. Would physiological serum levels of PA enhance vancomycin-mediated killing? It would be useful for the authors to comment on how host-derived PA might modulate vancomycin activity in vivo.

*Reviewer #3 (Recommendations for the authors):*

There is a lot of data showing that the combination causes effects that other peptidoglycan-targeting treatments do not, but formulating a specific mechanism could be aided by showing what other peptidoglycan-targeting treatments with known mechanisms do that this one does not, in order to eliminate those alternative mechanistic hypotheses.

Likewise, some ways to specifically enhance or lessen the effectiveness of the combination treatment based on the proposed mechanism would be persuasive to validate the mechanistic hypothesis, conceptually similar to how mutations that are consistent with a proposed mechanism in the effect they have on antibiotic efficacy help validate a protein as a target of an antibiotic.

If authors believe that the mechanism of the vancomycin-palmitoleic acid combination is conserved across all the species they investigate, then the larger *B. subtilis* cells might offer ways to probe the mechanism that the small S. aureus does not.

It seems that in several of the phenotypes that the authors monitor, additional insights could be gained by pretreating cells with one of the agents for varying lengths of time before adding another, to more precisely ascertain their individual contributions to the combined effect.

---

## [Author Response]

Essential revisions1. It is unclear if bacteria in other growth phases would be killed by dual PA and Vanc therapy. Please test if bacterial cultures approaching the stationary phase are also effectively killed by UFAs plus Vanc. Results will dictate how much can be concluded about tolerant cells.

We thank the reviewer for this suggestion. In the revised version of the manuscript, we have clarified the text in our methods section to indicate the growth phase and the length of time that the cultures are grown prior to antibiotic addition. We specifically utilized bacterial populations approaching stationary phase in all experiments in this manuscript, as indicated by the minimal increase in bacterial density from the time of challenge (T0) to the final time point evaluated (typically T6) (Figure 1B,C). To better evaluate tolerant populations, we chemically induced tolerance using arsenate to reduce ATP levels (Conlon et al., 2016) or CCCP which collapses the proton motive force. We found that palmitoleic acid still potentiates vancomycin killing against these tolerant populations of *S. aureus* (figure 1—figure supplement 3).

2. The proposed mechanism of UFA/Vanc synergy of accumulation of lipid II and alterations of membrane fluidity is partially demonstrated. Please strengthen these findings with the following experiments:– Assess lipid II buildup under treatment conditions employed and thus determine if there is a correlation between the degree of lipid II buildup and the extent of killing. Does the extent of lipid II buildup change over time under vancomycin/bacitracin treatment and if so, does the magnitude and/or rapidity of PA bactericidal effects vary depending on levels of accumulated lipid II at the time of PA addition?

We thank the reviewers for this critical suggestion. As these experiments are specialized, we formed a collaboration with Dr. Tanja Schneider’s group to measure membrane-bound cell wall intermediates following treatment with palmitoleic acid and vancomycin. Indeed, these assays showed that lipid II accumulates, within 30 minutes, in dual treated conditions, but unexpectedly, palmitoleic acid also led to the accumulation of the lipid anchor, bactoprenol (Figure 3). This is an important addition and we have revised our manuscript accordingly and now describe how the combined accumulation of bactoprenol, lipid II, and the insertion of palmitoleic acid into the membrane is driving RIF formation, protein delocalization, and cell death. We discuss these findings in lines 199-216 of the revised manuscript.

– Previous work has shown disruption/delocalisation of the divisome via treatment with daptomycin (Muller et al., 2016 PNAS) or zaragozic acid (Garcia-Fernandez et al., 2017 Cell). A floA mutant also has disrupted fluid domains and a crt mutant has a partial phenotype. Overexpression of GpsB antisense mRNA from a multicopy plasmid leads to membrane patches very similar to what is observed here (Eswara et al. eLife 38856 2018 Figures2 and 4). Based on the proposed mechanism here, test whether the effect of the PA treatment is abrogated in some of these conditions.

We appreciate the reviewers suggestions to evaluate conditions that have been linked to similar phenotypes as those observed in our study. Palmitoleic acid does not synergize with daptomycin (Figure 4 —figure supplement 5A), presumably because it is already a potent bactericidal agent that leads to RIF formation on its own. We also evaluated *floA*::Tn and *crtM*::Tn JE2 strains from the Nebraska Transposon Mutant Library. Interestingly, neither strain abrogated the synergy between palmitoleic acid and vancomycin (Figure 4 —figure supplement 5B-D). A recent study in *B. subtilis* showed that a *floAT* mutant increased the rigidity of the membrane overall, but local fluidity at the septum remained increased due to the presence of lipid II which innately disturbs membrane packing (Zielinska et al. 2020). It is possible that we do not see an impact on antibiotic susceptibility because the absence of flotillins does not impact fluidity at the septum. Furthermore, we confirmed that flotillins were not involved by pre-treating *S. aureus* with zaragozic acid, a statin that disrupts flotillin rafts, which had no impact on synergy (Figure 4 —figure supplement 5E). We have discussed these data in lines 287-301 of the revised manuscript.

– Figure 6 does not show equivalent experiments with fatty acids that do not synergise with vancomycin. Please provide this control.– Figure 4 shows that PA causes membrane fluid patches, which may or may not contribute to enhanced killing. Please test whether non-potentiating fatty acids cause patches. If PA specifically causes patches, it would provide stronger support of the model according to which these fluid domains contribute to killing.

We thank the reviewer for these suggestions. Supplements for figure 4 and 7 illustrate that the distinct phenotypes we observe with the palmitoleic acid-vancomycin combination are not present in *S. aureus* treated with a representative non-potentiating CMAA combined with vancomycin nor when cells are treated with a cell wall acting antibiotic that does not synergize with palmitoleic acid.

– Lines 280-282: 'Interestingly, these ultrastructural alterations of the cell membrane and septa parallel those reported for lipoglycopeptide treated *S. aureus*', implies that the combination of vancomycin and PA mimics lipoglycopeptide. Please test by combining a lipoglycopeptide with PA, which would presumably not synergise.

We are grateful for this reviewers suggestion and as expected oritavancin did not impact synergy between palmitoleic acid and vancomycin (figure 6 —figure supplement 1).

3. Figure 3b would benefit from a positive control to clarify the biological significance of the differences seen. What is the ATP loss with a membrane-damaging agent such as nisin?

We have since reevaluated our previous membrane permeability assays. Previously we used 7 μg/ml palmitoleic acid as that was the highest concentration, we could use without causing quenching the membrane potential dye DiSC_3_. However, to evaluate permeability more accurately we have since increased palmitoleic acid to the concentration used throughout the manuscript (11 μg/ml) and used SYTOX green which is not quenched at these concentrations. We have also included nisin as a positive control. We find that while palmitoleic acid alone increases ATP leakage, dual treated cells have a significant increase in small molecule leakage when compared to monotherapy, similar to nisin. However, using SYTOX green, we find that leakage is not due to pore formation as uptake of SYTOX green only occurred in nisin treated conditions. These data are presented in figure 5 and discussed in lines 333-353 of the revised manuscript.

4. Lines 205 and 206 state: 'Vancomycin induces palmitoleic acid incorporation at the septum leading to disruption of functionalized fluid regions'. However, the fluid patches shown in figure 4 are not associated with septa. For example, in Figure 4b, Van +PA(10), none of the patches appear to be near the septa. Please rephrase.

We appreciate the reviewer identifying this error, we have since rephrased that section in the revised manuscript.

5. Figure 1b, c. A t-test is inappropriate here because of the multiple comparisons (it is stated that no significant differences were detected unless indicated). Please use a 2-way repeated measures ANOVA. Figure 3b. Were stats done on 9 data points for each condition (3 independent replicates X 3 technical repeats?) as indicated by the figure? This would infer a much greater statistical power than is appropriate and increase the likelihood of P<0.05. Statistical testing should be done with the means of the 3 independent experiments.

We appreciate the reviewers identification of improper statistical analysis in our manuscript. We have since completed a two-way repeated measures ANOVA on the data in figure 1B,C and have clarified the figure legend (lines 132-135).

The data that was previously labeled as Figure 3B has now been repeated with different conditions and labeled as Figure 5A and we have performed one-way ANOVA with the means of the 2 independent experiments (lines 362-365). The methods and figure legend have been revised for clarity.

6. Lines 245-247 state: '…dual treated cells had a significant decrease in cell volume after 30 minutes (Figure 4F), suggesting that after prolonged exposure dual treated cells shrink, likely reflecting the leakiness of the membrane'. This set of experiments needs strengthening. Measure cells in each of the growth phases and determine whether this affects cell size. It appears that VAN-treated, and possibly PA-treated, cells have a greater volume than control. This would not fit with the conclusions in the text or figure legend stating that all conductions had p<0.0001 relative to control.

We thank the reviewer for their recommendation, we have since added supplement graphs of cell volume for each treatment condition and time point based on the cell phase (figure 4 —figure supplement 3 and figure supplement 4). These data illustrate that after 10 minutes, vancomycin monotherapy led to cells in phase 1 and 2 to yield a significant increase in cell volume, which is well reported in the literature (Salamaga et al. 2021). In comparison, palmitoleic acid monotherapy only resulted an increased cell volume for cells in phase 1. After 30 minutes of vancomycin or palmitoleic acid monotherapy, cells in both phase 1 and phase 2 had a significant increase in cell volume compared to the control cells. Interestingly, while palmitoleic acid monotherapy did not yield a significant difference in cell volume in phase 3 cells, vancomycin monotherapy resulted in a decrease in cell volume. These new data are now discussed in line (274-279) of the revised manuscript.